# Evaluating Posterior Probabilities:
# Decision Theory, Proper Scoring Rules, and Calibration

**Luciana Ferrer**
*lferrer@dc.uba.ar*
*Instituto de Ciencias de la Computación*
*Universidad de Buenos Aires - CONICET, Argentina*

**Daniel Ramos**
*daniel.ramos@uam.es*
*AUDIAS Lab. - Audio, Data Intelligence and Speech*
*Escuela Politécnica Superior, Universidad Autónoma de Madrid, Spain*

**Reviewed on OpenReview:** *https://openreview.net/forum?id=qbrEOLR7fF*

## Abstract

Most machine learning classifiers are designed to output posterior probabilities for the classes given the input sample. These probabilities may be used to make the categorical decision on the class of the sample; provided as input to a downstream system; or provided to a human for interpretation. Evaluating the quality of the posteriors generated by these system is an essential problem which was addressed decades ago with the invention of proper scoring rules (PSRs). Unfortunately, much of the recent machine learning literature uses calibration metrics—most commonly, the expected calibration error (ECE)—as a proxy to assess posterior performance. The problem with this approach is that calibration metrics reflect only one aspect of the quality of the posteriors, ignoring the discrimination performance. For this reason, we argue that calibration metrics should play no role in the assessment of posterior quality. Expected PSRs should instead be used for this job, preferably normalized for ease of interpretation. In this work, we first give a brief review of PSRs from a practical perspective, motivating their definition using Bayes decision theory. We discuss why expected PSRs provide a principled measure of the quality of a system's posteriors and why calibration metrics are not the right tool for this job. We argue that calibration metrics, while not useful for performance assessment, may be used as diagnostic tools during system development. With this purpose in mind, we discuss a simple and practical calibration metric, called calibration loss, derived from a decomposition of expected PSRs. We compare this metric with the ECE and with the expected score divergence calibration metric from the PSR literature and argue, using theoretical and empirical evidence, that calibration loss is superior to these two metrics.

## 1 Introduction

High-stakes machine learning applications, like those used to make health, military or legal decisions, often require systems that can provide a measure of uncertainty of the prediction given the input sample (Tomsett et al., 2020; Quinonero-Candela et al., 2005). In classification tasks, the uncertainty is a property of the posterior probability for the classes given the input sample. We use the term *probabilistic classifier* to refer to a classification system that outputs posterior probabilities. The posterior probabilities produced by a good probabilistic classifier can be used to make decisions for a given cost function using Bayes decision

theory (DeGroot, 1970; Bernardo & Smith, 1994; Jaynes, 2003). They can also be readily interpreted by an end user or passed on to a downstream system.

Evaluating the quality of the posteriors produced by a classification system is not a trivial task since, unlike for the evaluation of categorical decisions for which class labels are used as ground truth, there are no ground-truth posteriors against which to compare the system-generated posteriors. Except in simulations, the *true* posterior distribution is not available to us. All we ever have are models trained on data. Every model we develop for a given problem provides us with a new probabilistic classifier that does inference of the class of an input sample in the form of a posterior distribution. The focus of this work is how assess the goodness of such models.

The conclusions in this paper are based on one fundamental premise: that classification decisions should be made rationally and that, in the face of uncertainty, rational decisions are supported by Bayes decision theory (see, for example, Good, 1952; Peterson, 2009; Parmigiani & Inoue, 2009; Savage, 1972). Bayes decision theory provides a procedure for making decisions based on a given posterior distribution such that the expectation of a cost function of interest is minimized (Duda et al., 2001; Bishop, 2006; Hastie et al., 2001). Given this premise, a principled way to assess the quality of posterior probabilities is through proper scoring rules (PSR), which are constructed as the cost that results from making Bayes decisions using the posteriors under evaluation. A PSR measures the quality of the class posterior distribution for one specific input sample. To obtain a metric to assess the quality of a probabilistic classifier's output we take the expectation of the PSR (EPSR) over the data (Brümmer, 2010; Filho et al., 2023). For example, the cross-entropy, widely used as objective function to train classification systems, is the expectation of the PSR given by the negative logarithmic loss.

A defining property of PSRs is that their expectation with respect to a given reference probability distribution over the classes is minimized when the distribution under evaluation coincides with this reference distribution. Hence, a low PSR expectation indicates that the distribution under evaluation is close to the reference distribution with respect to which the expectation is taken. Note that we need to refer to a reference distribution since, as mentioned above, the true distribution is never available in practice.

The term PSR was first introduced by Winkler & Murphy (1968) who motivated their work by the need to assess the quality of weather forecasts. Brier score and the negative logarithmic loss (NLL), proposed in the 1950s, are special cases of PSRs (Brier, 1950; Good, 1952). PSRs were later further studied in a large number of works (see, for example, Dawid et al., 2016; Gneiting & Raftery, 2007; Bröcker, 2009; Ovcharov, 2015; Gneiting, 2011, among many others).

In contrast to this large body of literature on PSRs, many of the recent machine learning works concerned with the evaluation of posteriors do not use PSRs for the task, resorting instead to calibration metrics (see, for example, Guo et al., 2017; Widmann et al., 2019; Gruber & Buettner, 2022; Nixon et al., 2019; Popordanoska et al., 2022; Vaicenavicius et al., 2019; Van Hoorde et al., 2015; Huang et al., 2020; Müller et al., 2019; Mukhoti et al., 2020; Minderer et al., 2021; Jiang et al., 2021; Desai & Durrett, 2020; Dehghani et al., 2023). A classification system is said to be well-calibrated if its output, $\mathbf{q}$, coincides with a reference posterior distribution for the classes given $\mathbf{q}$, for every possible input sample, $x$ (Bröcker, 2009; Guo et al., 2017; Widmann et al., 2019). As before, we need to refer to a reference posterior since the true posterior is not available in practice. The reference posterior needed to check calibration is yet another model of the posterior, one that we trust to be good.

Note that the calibration definition does not refer to posteriors for the classes *given the input sample.* Instead, it refers to posteriors for the classes *given the system's output.* Hence, good calibration does not imply that the system's posteriors are doing a good job of inferring the classes from the inputs, which is what matters in practice. In fact, a calibrated system can be useless. For example, a naive system that always outputs the prior probabilities of the classes is perfectly calibrated but does not provide any information about the input samples. As we will see, the overall quality of the system's posteriors, as measured by EPSRs, can be decomposed in two terms: a calibration and a discrimination or refinement component (Filho et al., 2023; Bröcker, 2009). The discrimination component quantifies the amount of information present in the system's output about the class of the samples while the calibration component quantifies the similarity between the

system's output and a reference distribution for the class given that output. Neither component by its own fully describes the goodness of the posteriors produced by the system.

As a consequence of the fact that calibration metrics only reflect one aspect of the posteriors' performance, they do not inform how useful the classifier will be in practice. To address this problem, papers that use calibration metrics resort to a separate metric, usually accuracy, error rate, or area under the ROC curve, to complement the calibration metric (see, for example, Guo et al., 2017; Van Hoorde et al., 2015; Minderer et al., 2021; Mukhoti et al., 2020). This is a problematic practice since it does not allow for a direct comparison between two systems. If a system's calibration error is lower but the error rate is higher than for another system, which of them is better? Which one should we choose for deployment if our main goal is to produce good posteriors for interpretation and decision-making? Also, this practice does not solve the problem of assessing whether the performance of the posteriors is good enough for a certain task. These questions are answered by EPSRs which provide a comprehensive measure of the value provided by the system's posteriors. Using calibration metrics, on the other hand, leads to unnecessary conflict.

Papers that report calibration metrics motivate its use by arguing that good calibration is an essential characteristic for a probabilistic system to be interpretable, safe, or reliable (Guo et al., 2017; Widmann et al., 2019; Mukhoti et al., 2020; Kumar et al., 2019; Minderer et al., 2021; Nixon et al., 2019), using statements like: "a network should provide a calibrated confidence" (Guo et al., 2017), "miscalibration [...] makes [...] predictions hard to rely on" (Mukhoti et al., 2020), and "model calibration is essential for the safe application of neural networks" (Minderer et al., 2021). The implicit or explicit implication in those statements is that a very discriminative system with relatively high calibration loss should not be used in high-stakes scenarios. We challenge this view and propose that calibration is neither necessary, nor sufficient for posterior probabilities to be useful to the end user. A system may be somewhat miscalibrated but still be useful, as long as the posteriors that it outputs result in sufficiently good Bayes decisions. In particular, a miscalibrated system may be much more useful than the perfectly-calibrated naive system mentioned above. If a system's EPSR is lower than the EPSR of an alternative system, we can conclude that its posteriors are better, regardless of the calibration error of either system. Hence, when evaluating the utility of posteriors we should not be concerned with whether they are calibrated or not. Assessing the quality of probabilistic classifiers is exactly the purpose of EPSRs. When the goal is to assess the value of an individual classifier or compare classifiers with each other to select the best one, there is no need to resort to the concept of calibration.

In this paper, we argue that the only purpose of calibration metrics should be to diagnose whether a system is well-calibrated in order to fix it if that is not the case, much like learning curves over validation and training data are used to diagnose overfitting. While regularization, early stopping, or smaller models may be explored when overfitting is detected, various approaches can be explored when miscalibration is detected. In particular, a miscalibrated system can be very easily improved by adding a post-hoc calibration stage (Filho et al., 2023): a transformation of the system output designed to reduce the calibration error. Such a stage, if successful, would result in a new system with a lower EPSR, which is our final goal. Beyond the use as a diagnostic tool during development, calibration metrics should not play any role in system evaluation since they do not provide any particular insight about the value of a system for the end user. If changing the system is not an option, assessing and reporting calibration performance has no practical role.

For the calibration analysis needed during system development, we propose to use the calibration loss metric. Calibration loss is obtained from a decomposition of an EPSR into calibration and discrimination terms, and directly reflects the improvement we would obtain if a post-hoc calibration stage was added to the system. A particular version of this metric was introduced in the literature years ago for the binary task of speaker verification (Brümmer & du Preez, 2006) and later adopted in forensic science (Ramos & Gonzalez-Rodriguez, 2013; Ramos et al., 2017; 2020). We compare it, theoretically and empirically, with the widely-used expected calibration error (ECE) metric (Naeini et al., 2015a; Guo et al., 2017), and argue that ECE has no theoretical or practical advantage over calibration loss having, on the other hand, various disadvantages.

The rest of this paper gives an introduction to Bayes decision theory and PSRs, highlighting their tight relationship. Then, it describes the problem of calibration, introduces the calibration loss metric, and compares it with the ECE and the expected score divergence, a classic calibration metric from the statistics

literature, providing novel insights and discussing the advantages of the calibration loss over those classic alternatives. Finally, it presents empirical results on synthetic and real datasets to demonstrate and further discuss the theoretical ideas in the previous sections. A substantial part of the content in this paper revisits well-established concepts, some dating back decades. Yet, we believe a discussion of these ideas is still needed in the machine learning community, given the current wide-spread use of calibration metrics as a proxy to assess the quality of a system's posteriors.

In summary, the goals of this work are to argue: 1) that calibration metrics should only be used during system development, for example, for the purpose of deciding whether a post-hoc calibration stage is needed in the system, 2) that for those cases, calibration loss, a simple and principled calibration metric, is preferable to the ECE and to the expected score divergence, 3) that the goodness of a probabilistic system, as will be perceived by an end user, a downstream system, or a decision stage, should be assessed using EPSRs, not calibration metrics.

## 2  From Bayes decision theory to calibration

The goal of this section is to review known concepts from the statistical learning literature and discuss them in the light of current trends in the machine learning literature, where the quality of posterior probabilities is most often assessed using the ECE or one of its variants. First, we discuss the reference distribution, a necessary construct when dealing with real data where the underlying data-generating distribution is not known. Then, we describe how to make optimal (Bayes) decisions for a given cost function selected for the application of interest. We then define PSRs as the cost of Bayes decisions motivating their use as metrics to evaluate the quality of posteriors. We also explain how they can be normalized to obtain interpretable metrics. Further, we explain how EPSRs can be understood as integrals over a family of Bayes risks, providing further intuition on why and how EPSRs reflect the quality of posteriors. Finally, we show how to decompose an EPSR into calibration and discrimination components in two ways, using the traditional divergence-entropy decomposition, and using the calibration loss decomposition, and compare the resulting metrics with the widely used ECE metric.

While most of the content in this section is based on decades-old concepts from the statistical literature, our treatment is different from that in most works in that the need for a reference distribution is made explicit in every step of the way. As we will see, this uncovers some practical issues that are usually not considered in the literature.

### 2.1  The reference distribution

In statistics and machine learning literature, it is common practice to explicitly or implicitly rely on 'the true distribution of the data'. We find this an ill-defined concept, which is not useful in practice and which, in particular, poses formidable obstacles to understanding calibration. Even if one could theoretically consider that a given dataset was created by sampling from a true underlying probability distribution, such a distribution will never be known to us—except in simulations.

In machine learning, though, both at training time and test time, it *is* very useful to work with probability distributions for[1] the data. By 'the data' we refer to some future, or otherwise unseen data, and by 'distributions for the data' we refer to probabilistic *predictions* of the unseen data. For example, if you are training a classifier by minimizing the cross-entropy over a supervised training dataset, you are effectively minimizing a prediction of the cross-entropy on future, unseen data. If you are testing your classifier with error-rate or cross-entropy, you are also effectively predicting how it will perform on future data. We can obtain the probabilistic predictions of the data, which we will call *reference* distributions, by selecting modeling assumptions and then fitting the model to a given dataset. Different assumptions will lead to different reference distributions.

---

[1]We follow the advice by Jaynes (2003) of using the preposition *for*, rather than *of* to refer to the relationships between distributions and data. Distributions that we choose to use to model some unseen data are not an intrinsic property of the unseen data, rather they are induced by assumptions and given data.

The reference distributions can be very simple. Perhaps the simplest reference distribution is the empirical one, obtained by uniformly sampling with replacement from a dataset. Expectations with respect to the empirical distribution are given by averages over the samples in the dataset. The empirical distribution is the most common reference distribution used in machine learning both for training and testing models, where cross-entropy or error rates are computed as averages over the training or test samples. As we will see, though, it turns out that empirical distributions *do not* provide useful references against which to judge calibration—if we make use of the classical definition of calibration. It is therefore necessary for our exploration and understanding of calibration to consider more general reference distributions than the empirical ones. In the theoretical sections below we will simply leave the reference undefined, assuming enough regularity conditions are imposed in the model to result in a good predictor of future data. We will discuss a simple way to build such reference distributions in Section 3.4.

## 2.2 Bayes decision theory

Assume that we have selected a cost function for our classification problem of interest, $C(h,d)$, where $h \in \mathcal{H} = \{H_1, \ldots, H_K\}$ is the true class of the sample, $d \in \mathcal{D}$ is the decision made by the system, and $C : \mathcal{H} \times \mathcal{D} \to \mathbb{R}$. Decisions can be categorical, in which case we take $\mathcal{D} = \{D_1, \ldots, D_M\}$. The set of decisions and the set of classes do not need to be the same. The decisions are, in general, the actions that will be taken based on the system's output (Duda et al., 2001). For example, $\mathcal{D}$ could include a "reject" or "abstain" option (see, for example, Bishop, 2006, section 1.5). Decisions can also be *soft*, in the form of a distribution over classes, in which case $\mathcal{D} = \mathbb{S}^K$, the simplex where $K$-class categorical distributions live.

Given the cost function, our goal will be to make optimal decisions in the sense that they minimize the expectation of this cost (Peterson, 2009), $\mathbb{E}_{h \sim P_r(h|x)}[C(h, d(x))]$, where we have explicitly added the dependency of $d$ on the input sample, $x$. The expectation is taken with respect to $P_r(h \mid x)$, a reference distribution for the class label given the input. This expectation is minimized by taking[2] (Bishop, 2006; Hastie et al., 2001):

$$d(x) = d_B(x) := \arg\min_d \sum_{i=1}^{K} C(H_i, d) P_r(H_i|x) \tag{1}$$

The decision $d_B$ is called the *Bayes decision*. If decisions are made in this way for every $x$, then the expected cost with respect to the joint distribution $P_r(h, x)$ is also minimized.

## 2.3 Proper scoring rules

Proper scoring rules (PSRs) are a family of functions specifically designed to assess the quality of posterior probabilities (Gneiting & Raftery, 2007; Brümmer, 2010). The principle behind PSRs is that the quality of posteriors is given by the quality of the Bayes decisions made with them: better decisions imply better posteriors. Formally, given a posterior for a sample $x$ provided by a classifier, which we denote[3] as $\mathbf{q}(x) = P_c(.|x)$, and a cost function $C(h, d)$, we can construct a PSR, $C^*(h, \mathbf{q})$, as follows (Dawid & Musio, 2014; Brümmer, 2010):

$$C^*(h, \mathbf{q}) = C(h, d_B(\mathbf{q})). \tag{2}$$

That is, $C^*$ is the cost of the Bayes decision made with $\mathbf{q}$. Note that here we have expressed $d_B$ as a function of $\mathbf{q}$ rather than $x$ since, as shown in Equation (1), Bayes decisions only depend on the posterior which, here, is given by $\mathbf{q}$.

The $C^*$ constructed in this way satisfies the defining property of PSRs which is that their expected value with respect to any distribution $\mathbf{p}$ over the classes is minimized if $\mathbf{q}$ coincides with $\mathbf{p}$ (Dawid & Musio, 2014; Brümmer, 2010). That is,

$$\mathbf{p} \in \arg\min_{\mathbf{q}} \mathbb{E}_{h \sim \mathbf{p}}[C^*(h, \mathbf{q})]. \tag{3}$$

---

[2]There may sometimes be more than one minimizing decision. In such cases we assume arg min applies a tie-breaker to return a single minimizing decision. The details of such tie breakers is unimportant here.

[3]We use bold $\mathbf{q}$ for vectors in $\mathbb{S}^K$ representing discrete distributions, and italic with a subindex $q_i$ to refer to the $i$th component of the $\mathbf{q}$ vector.

When the minimum is unique, the PSR is called strict.

A PSR measures the quality of the posterior vector $\mathbf{q}$ for a single sample. To obtain a metric that can be used for evaluation, we can compute the expectation of the PSR with respect to the joint reference distribution:

$$\text{EPSR} = \mathbb{E}_{P_r(h,x)}\left[C^*(h, \mathbf{q}(x))\right]. \tag{4}$$

Now, since $C^*$ is a function of $x$ only through $\mathbf{q}$, we can invoke the law of the unconscious statistician (DeGroot & Schervish, 2014, page 213) and compute the expectation with respect to $\mathbf{q}$ instead:

$$\text{EPSR} = \mathbb{E}_{P_r(h,\mathbf{q})}\left[C^*(h, \mathbf{q})\right]. \tag{5}$$

This equation shows that we do not need a reference distribution for $(h, x)$ to compute the EPSR. Instead, we only need a reference for $(h, \mathbf{q})$.

In practice, to compute this metric, we take the reference distribution to be the empirical distribution in the test dataset so that the EPSR is given by

$$\text{EPSR} = \frac{1}{N}\sum_{t=1}^{N} C^*(h_t, \mathbf{q}_t). \tag{6}$$

where $q_t$ and $h_t$ are the classifier's output and the label for sample $t$ in a test dataset containing $N$ samples.

Different EPSRs can be obtained by using different cost functions $C$. Below we will discuss three of the most common EPSRs.

### 2.3.1 Bayes risk

When the decision $d$ is categorical ($d \in \{D_1, \ldots, D_M\}$), the cost function can be expressed as a matrix of costs $c_{ij}$ for each combination of true class $H_i$ and decision $D_j$. The expected cost with respect to the empirical distribution in a test dataset is given by the average cost over the samples:

$$\text{EC} = \sum_{i=1}^{K}\sum_{j=1}^{M} c_{ij}\frac{N_{ij}}{N} = \sum_{i=1}^{K}\sum_{j=1}^{M} c_{ij}P_i R_{ij} \tag{7}$$

where $P_i = N_i/N$ is the empirical prior probability of class $H_i$, $N_i$ is the number of samples of class $H_i$, and $R_{ij} = N_{ij}/N_i$ is the fraction of samples from class $H_i$ for which the system made decision $D_j$. The EC can be normalized to ease interpretation by dividing it by the EC of a system that outputs always the least-costly decision, which is given by $\min_d \sum_{i=1}^{K} c_{id}P_i$. A special case of the EC is obtained for the 0-1 cost matrix where $c_{ij} = 1$ when $i \neq j$ (the decision is incorrect), and $c_{ij} = 0$ when $i = j$ (the decision is correct). The average cost in this case reduces to the standard error rate, which is equal to one minus the accuracy.

The EC can be computed regardless of how decisions are made. If decisions are made using Bayes decision theory, though, the resulting EC is an EPSR commonly called Bayes risk (Duda et al., 2001). Given a cost matrix, Equation (1) corresponds to a specific partition of the simplex $\mathbb{S}^K$ into decision regions. For binary classification with a square cost matrix ($M = 2$), the regions can be determined by a threshold on either of the two posteriors. For the 0-1 cost, the Bayes decision is the class with the maximum posterior (Hastie et al., 2001), or argmax decision. Hence, the error rate of argmax decisions, a widely-used metric in the machine learning classification literature, is one instance of the Bayes risk. The left plot in Figure 1 illustrates a cost function for binary classification when the Bayes threshold is used.

When evaluating Bayes risk, we are, as for any EPSR, measuring the quality of the posteriors used to make the Bayes decisions. Yet, we are restricting our assessment of the posteriors to one specific operating point defined by the cost matrix. As a result of this, the Bayes risk is not a strict PSR: Two classifiers that produce posterior distributions for which the Bayes decisions for the selected cost matrix are the same, will lead to the same Bayes risk. The Bayes risk for other cost matrices, though, will not necessarily be the same for both classifiers, since their posteriors are different.

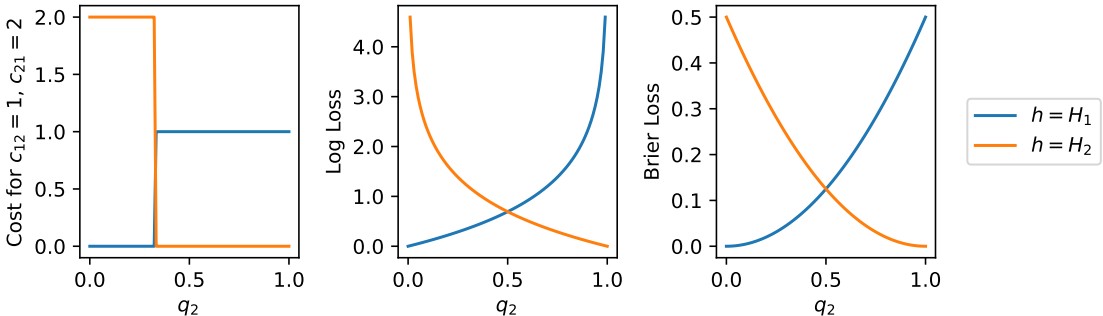

Figure 1: Three PSRs on a binary classification task as a function of the posterior for class $H_2$, $q_2$: (left) the cost function with $c_{12} = 1$ and $c_{21} = 2$ for the Bayes threshold (given by $c_{12}/(c_{12} + c_{21})$), (center) the logarithmic loss, and (right) the Brier loss. The blue curve corresponds to the loss for samples of class $H_1$, which is maximum when $q_2 = 1$ (conversely for the orange curve). The logarithmic loss goes to infinity in the extremes in which the posterior is 1 for the wrong class, while the other two losses stay bounded. The expectation of these PSRs corresponds to the Bayes risk, the Cross-entropy, and the Brier score, respectively.

### 2.3.2 Cross-entropy

When we want to evaluate the quality of posteriors in general rather than for a single operating point, we need to resort to strict PSRs. One such PSR can be obtained by setting $C$ to be the negative logarithmic loss (NLL), $C(h, \mathbf{d}) = -\log(d_h)$, where $\mathbf{d} \in \mathbb{S}^K$ and, in a slight abuse of notation, $d_h$ is the element of $\mathbf{d}$ corresponding to class $h$ (if $h = H_i$, then $d_h$ is the $i$th element of vector $\mathbf{d}$, also denoted as $d_i$). It can be shown that, for this cost function, the Bayes decision (Equation 1) is $d_B(\mathbf{q}) = \mathbf{q}$, since, by Gibbs inequality, $\sum_i C(H_i, \mathbf{d}) \, q_i = -\sum_i \log(d_i) \, q_i \geq -\sum_i \log(q_i) \, q_i$. That is, the (soft) decision that minimizes the expected NLL is $\mathbf{q}$ itself. Hence, the PSR resulting from this cost function is $C^*(h, \mathbf{q}) = -\log(q_h)$ (Equation 2), which is again the NLL. It is easy to show that the NLL is a strict PSR (Dawid & Musio, 2014). The middle plot in Figure 1 illustrates the logarithmic loss for binary classification.

The expectation of the NLL over the data is the cross-entropy, which is widely used as objective function for training. Taking the expectation with respect to the empirical distribution, we get

$$\text{CE} = -\frac{1}{N} \sum_{t=1}^{N} \log(q_{h_t}(x_t)), \tag{8}$$

where $q_{h_t}(x_t)$ is the system's output for sample $x_t$ for class $h_t$, the true class of sample $t$. This expression can be rewritten to make its dependence on the class priors explicit (see Appendix A), which allows us to manipulate these priors independently of those in the test data. This is useful when the priors in our test data do not reflect those we expect to see when the system is deployed, in which case we can use the target priors instead of the ones in our data when computing the CE.

The absolute CE values are not easily interpretable. This issue, though, can be solved by normalizing its value with the CE of the best naive system. A naive system is one that does not have access to the input data. The best naive system for any EPSR is the one that outputs the prior distribution in the test data (Brümmer, 2010, Section 2.4.1). The CE of a system that always outputs the prior distribution is the entropy of such distribution, $-\sum_{i=1}^{K} P_i \log(P_i)$. Dividing the CE by this value we obtain the normalized CE, or NCE for short. The NCE values can be readily interpreted: values above 1.0 mean that the system is worse than the best naive system and, hence, one should either 1) throw it away and replace it with a system that outputs the prior distribution, or 2) fix it by doing calibration (as we will see in Section 2.4.2). A well-calibrated system will never have a normalized CE larger than one.

### 2.3.3 Brier score

If we take $C(h, \mathbf{d}) = \frac{1}{K} \sum_{i=1}^{K} (d_i - I(h = H_i))^2$, with $\mathbf{d} \in \mathbb{S}^K$, we get another PSR called Brier loss. Here, $I(h = H_i)$ is the indicator function which is 1 if the argument is true and 0 otherwise. As for the NLL, it can be shown that the Bayes decision for this loss is the posterior used to make the decision. Hence, as for the NLL, the PSR coincides with the cost function. The right plot in Figure 1 illustrates the behavior of the Brier loss for binary classification.

Averaging the Brier loss over the data we get the Brier score

$$\text{BS} = \frac{1}{N} \sum_{t=1}^{N} \frac{1}{K} \sum_{i=1}^{K} (q_i(x_t) - I(h_t = H_i))^2. \tag{9}$$

As for the CE, this expression can be manipulated to show its dependency with the priors explicitly which allows us to turn the priors into parameters of the metric (see Appendix A). Further, as for CE, it can be shown that BS is a strict PSR (Dawid & Musio, 2014). Unlike the CE, though, which can be infinite if the posterior for the right class is 0.0 for any sample in the dataset, BS is bounded, being more forgiving of extremely incorrect posteriors (see Figure 1). Finally, we can compute a normalized version of BS, NBS, by dividing its value by the BS of the best naive system, which is given by $\sum_{i=1}^{K} P_i(1 - P_i)/K$.

### 2.3.4 EPSRs as integrals over Bayes risks

As explained, for example, by Gneiting & Raftery (2007) and by Brümmer (2010), we can construct a PSR, $C_W^*$, by taking the following integral:

$$C_W^*(h, \mathbf{q}) = \int_{\mathbb{S}^K} W(\mathbf{a}) \, C_{\mathbf{a}}^*(h, \mathbf{q}) \, d\mathbf{a} \tag{10}$$

where $h$ is the class of the sample, $\mathbf{q} \in \mathbb{S}^K$ is a posterior distribution, and $W(\mathbf{a})$ is a function of $\mathbf{a} \in \mathbb{S}^K$ that satisfies $W(\mathbf{a}) \geq 0 \; \forall \mathbf{a}$. The $C_{\mathbf{a}}^*$ function inside that integral is the PSR obtained as explained in Section 2.3.1 for the following cost function:

$$C_{\mathbf{a}}(i, j) = \frac{1 - I(i = j)}{(N - 1) \, a_i}, \tag{11}$$

where $C_{\mathbf{a}}(i, j)$ is the cost for deciding $D_j$ when the true class of the sample is $H_i$. For this construction, we take the set of decisions to be the same as the set of classes. This matrix has zeroes in the diagonal and a cost proportional to $1/a_i$ everywhere else. Taking the expectation with respect to the data on both sides of Equation (10), we get that

$$\text{EPSR} = \int_{\mathbb{S}^K} W(\mathbf{a}) \, \text{Risk}(\mathbf{a}) \, d\mathbf{a} \qquad \text{where} \qquad \text{Risk}(\mathbf{a}) = \mathbb{E}\left[C_{\mathbf{a}}^*(h, \mathbf{q})\right]. \tag{12}$$

The function $W(\mathbf{a})$ determines how much each Bayes risk inside the integral influences the final value. Taking $W(\mathbf{a})$ to be uniform in the simplex, we obtain the CE. For the binary case, taking $W(\mathbf{a})$ to be a Beta distribution with both parameters equal to 2, so that $W(\mathbf{a}) = a_1 \, (1 - a_1)/B(2, 2)$, where $B$ is the beta function, we obtain the BS. The proofs for these statements can be found in (Brümmer, 2010, section 7.4). Section 3.1 shows an illustration of this way of constructing the CE and BS metrics.

### 2.4 Calibration

Following the literature (for example, Bröcker, 2009; Guo et al., 2017; Vaicenavicius et al., 2019; Nixon et al., 2019; Gruber & Buettner, 2022), a classifier, $P_c$, that for input $x$, outputs the class posterior, $\mathbf{q} = [q_1, \dots, q_K]$, where $q_i = P_c(H_i \mid x)$, has *perfect calibration* with respect to a reference distribution $P_r$, if:

$$q_i = P_r(H_i \mid \mathbf{q}), \; \forall i, \mathbf{q} \tag{13}$$

This notion of calibration is sometimes called *canonical*, *distribution* or *strong* calibration (Widmann et al., 2019; Gopalan et al., 2024; Popordanoska et al., 2022). Weaker definitions of calibration have been proposed, like class-wise and confidence calibration (Kull et al., 2019), which refer to a single value from the posterior distribution: the value for each of the classes, or the largest value (usually called confidence), respectively. These weaker notions were proposed because they are easier to verify empirically, since the conditioning on a single value instead of the full posterior vector simplifies the problem. An example of this will be discussed in Section 2.5.2 on the expected calibration loss, a metric which, for the multi-class case, evaluates confidence calibration. Unfortunately, these weaker notions of calibration, while easier to check, are often very poor proxies for the strong notion, as discussed in prior works (Nixon et al., 2019; Popordanoska et al., 2022). In this work, we focus on the strong notion of calibration which, as explained in Section 2.4.1, guarantees the optimality of Bayes decisions for the given classifier. We believe weaker notions are unnecessary since, as we argue extensively throughout this work, the goal of assessing calibration should only be to decide whether adding a calibration stage would improve the quality of the posteriors and this can be done using the calibration loss metric without the need to weaken the definition. The complexity in measuring calibration is not in the definition but rather in determining the reference distribution.

In most of the literature cited above, the role of $P_r$ is implicitly deferred to the ill-defined concept of the 'true distribution'. With the reference made explicit, we note that a classifier may be calibrated with respect to some reference distribution, but miscalibrated with respect to another. In this section we carefully analyze this definition of perfect calibration and also compare it to the optimal classifier deferring the discussion of practical ways to define that reference to Section 2.4.2.

Given a reference distribution for $h$ and $x$, $P_r(h, x)$, the *optimal classifier* is the one that outputs the class posterior, $\mathbf{p} = [p_1, \ldots, p_K] = P_r(\cdot \mid x)$ with components:

$$p_i = P_r(H_i \mid x) = \frac{P_r(H_i, x)}{\sum_{k=1}^{K} P_r(H_k, x)} \tag{14}$$

Note that this distribution is not the reference for perfect calibration of the classifier: the right hand side in Equation (13) is $P_r(. \mid \mathbf{q})$, not $P_r(. \mid x)$. In other words, perfect calibration does not ensure that $\mathbf{q}$ is the optimal classifier—it is a weaker condition that can hold for $\mathbf{q} \neq \mathbf{p}$, that is, for a suboptimal classifier.

Before moving on to analyzing the calibration of $\mathbf{q}$, we need to understand the implicit perfect calibration of the optimal classifier, $\mathbf{p}$. Since $\mathbf{p}$ is a function of $x$, it is also a random variable for which the joint, marginal and conditional distributions can be derived from $P_r(h, x)$. The perfect calibration of $\mathbf{p}$ is defined as $p_i = P_r(H_i \mid \mathbf{p})$. This equality always holds if $\mathbf{p}$ is the optimal classifier given by Equation (14). This can be seen as follows:[4]

$$\begin{aligned} P_r(H_i, \mathbf{p}) &= \int_{\tilde{x}: P_r(\cdot \mid \tilde{x}) = \mathbf{p}} P_r(H_i, \tilde{x}) \, d\tilde{x} \\ &= \int_{\tilde{x}: P_r(\cdot \mid \tilde{x}) = \mathbf{p}} P_r(H_i \mid \tilde{x}) P_r(\tilde{x}) \, d\tilde{x} \\ &= p_i \int_{\tilde{x}: P_r(\cdot \mid \tilde{x}) = \mathbf{p}} P_r(\tilde{x}) \, d\tilde{x} \\ &= p_i \, P_r(\mathbf{p}) \end{aligned} \tag{15}$$

where we used that $P(H_i \mid \tilde{x})$ is *constant* with value $p_i = P_r(H_i \mid x)$, for the values of $\tilde{x}$ over which we are integrating. Rearranging, we find:

$$P_r(H_i \mid \mathbf{p}) = \frac{P_r(H_i, \mathbf{p})}{P_r(\mathbf{p})} = p_i = P_r(H_i \mid x) \tag{16}$$

This result can be summarized succinctly as:

$$P_r(\cdot \mid \mathbf{p}) = \mathbf{p} = P_r(\cdot \mid x) \tag{17}$$

---

[4] A measure-theoretic (terse, perhaps less accessible) derivation is given in the appendix of Bröcker (2009). This derivation, proposed by Niko Brümmer during personal communications with the authors, is more similar to a derivation of likelihood-ratio calibration, in the appendix of Slooten & Meester (2012).

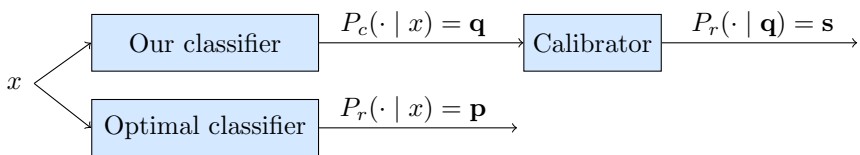

Figure 2: Diagram of the posteriors in Section 2.4. The posterior $\mathbf{q}$ is the output of our classifier. After calibration according to the reference distribution $P_r(h, x)$, we obtain the posterior $\mathbf{s}$. If our classifier is perfectly calibrated with respect to the reference, then $\mathbf{s} = \mathbf{q}$. On the other hand, the optimal classifier under the reference distribution $P_r(h, x)$ is the one that produces $\mathbf{p}$. These are the posteriors we ideally want to have. In general, though, $\mathbf{q}$ is different from $\mathbf{p}$. Further, perfect calibration does not guarantee that $\mathbf{s}$ is equal to $\mathbf{p}$.

This is intuitive: We have already inferred $h$ from $x$ in the form of $\mathbf{p}$, so that if we want to infer $h$ directly from $\mathbf{p}$, without any extra information, the result remains the same.

We have shown that $\mathbf{p}$, the optimal classifier for a reference $P_r(h, x)$, is perfectly calibrated with respect to the posterior distribution consistent with that reference. Now, recall that our classifier is the model $P_c$, that outputs the posterior $\mathbf{q} = P_c(\cdot \mid x)$, with components $q_i = P_c(H_i \mid x)$. In general, our classifier will be different from the optimal classifier:

$$\mathbf{q} = P_c(\cdot \mid x) \neq \mathbf{p} = P_r(\cdot \mid x). \tag{18}$$

Here and elsewhere, by '$\neq$', we mean *not equal in general*. In contrast to $P_r(h, x)$ which provides the full joint distribution, all we need to practically implement a classifier is the conditional, $P_c(h \mid x)$. We can however allow the thought experiment to extend this model to $P_c(h, x) = P_c(x)P_c(h \mid x)$. Then Equation (15) shows that the classifier is also perfectly calibrated—if we use $P_c$ itself as reference:

$$P_c(\cdot \mid \mathbf{q}) = \mathbf{q} = P_c(\cdot \mid x) \tag{19}$$

What we want to do however, is to judge the calibration of $\mathbf{q}$ with respect to $P_r$ instead. Again, since $\mathbf{q}$ is a function of $x$, the model $P_r$ provides also the posterior $P_r(. \mid \mathbf{q})$, which we will call $\mathbf{s}$, against which $\mathbf{q}$ may be judged.[5] We have now defined a total of three different posteriors, $\mathbf{q} \neq \mathbf{s} \neq \mathbf{p}$: our classifier, the class posterior given our classifier's output, and the optimal classifier, the latter two derived from the reference distribution. Their definitions and relationships may be summarized as (see also Figure 2):

$$\mathbf{q} = P_c(\cdot \mid x) = P_c(\cdot \mid \mathbf{q}) \neq \mathbf{s} = P_r(\cdot \mid \mathbf{q}) \neq P_r(\cdot \mid \mathbf{p}) = \mathbf{p} = P_r(\cdot \mid x) \tag{20}$$

The first '$\neq$' is simply because $P_c \neq P_r$. The second '$\neq$' follows from the *data processing inequality*[6] if we assume the function from $x$ to $\mathbf{q}$ is non-invertible. As noted above, the literature defines *perfect calibration* as the special case when $\mathbf{q} = P_r(\cdot \mid \mathbf{q})$, that is, when the first '$\neq$' in Equation (20) is replaced with equality. Even then, we still have the second '$\neq$', which highlights the fact that a perfectly calibrated classifier is not necessarily optimal.

The classifier $P_c$ is *optimal* with respect to the reference, if $P_c(\cdot \mid x) = \mathbf{q} = \mathbf{p} = P_r(\cdot \mid x)$. But this is a much stronger requirement than merely having perfect calibration, as defined by Equation (13). While $\mathbf{q} = \mathbf{p}$ for all $x$ implies perfect calibration by Equation (17), the converse is not true. An extreme counterexample is the naive classifier, say $P_c = P_0$, that completely ignores its input, while having perfect calibration:

$$q_i = P_0(H_i \mid x) = P_r(H_i) = P_r(H_i \mid \mathbf{q}) \tag{21}$$

where the first two equalities define $P_0$: it outputs the class prior as given by the reference $P_r$, irrespective of the value of $x$. The last equality, which shows that $P_0$ has perfect calibration, follows because $\mathbf{q}$ is constant.

---

[5] $P_r(h \mid \mathbf{q}) \propto P_r(h, \mathbf{q}) = \int_{\tilde{x}:P_c(\cdot\mid\tilde{x})=\mathbf{q}} P_r(h, \tilde{x}) \, d\tilde{x}$.

[6] With $P_r$ as reference, the data processing inequality states that $I(h; x) = I(h; \mathbf{p}) \geq I(h; \mathbf{q})$, where $I$ denotes mutual information (Cover & Thomas, 2006). While $\mathbf{p}$ contains all of the information about $h$ that is present in $x$, $\mathbf{q}$ generally contains less. The non-invertible function $x \mapsto \mathbf{p}$ is also subject to the data processing inequality, but in this special case, equality is given by Equation (15).

This is just an example of a perfectly-calibrated but useless classifier that highlights the problem with using calibration metrics to assess performance of probabilistic classifiers.

To understand why calibration metrics should not be used to assess posterior performance, we may also consider an analogy between a system's calibration quality and the size of its training set. These two aspects are analogous in the sense that both *affect* but do not *reflect* the quality of the posteriors produced by the system. Specifically, a system may be trained with a small amount of data or be miscalibrated and still produce good posteriors. Conversely, a system may be trained with a large amount of data and be perfectly calibrated and still produce poor posteriors. Further, if we increase the size of the training set for a given system, the quality of the system will usually improve or, at least, not degrade. Similarly, if we calibrate a given system (as explained in Section 2.4.2), it will usually improve or, at least, not degrade. Finally, given two different systems, the size of their training sets or its calibration quality are not reliable indicators of which system has better posteriors. One system may be trained with more data or be better calibrated than another but produce poorer posteriors than the other. So, just as we would not use the size of the training set as a measure of the quality of a system's posteriors, we should not use calibration metrics for this purpose either. Instead, EPSRs are the right tool for that job.

### 2.4.1 Calibration and Bayes decisions

Most machine learning classifiers make their final decisions based only on the classifier output, $\mathbf{q}$. This means that the $x$ in Equation (1) is given by $\mathbf{q}$, since those are the input features available for decision making. Replacing $x$ with $\mathbf{q}$ in Equation (1), we get that the Bayes decisions based on $\mathbf{q}$ are given by

$$d_B(\mathbf{q}) = \arg\min_d \sum_{i=1}^{K} C(H_i, d) P_r(H_i \mid \mathbf{q}) \tag{22}$$

Those decisions minimize the expected cost with respect to that same reference distribution, $P_r$.

According to the definition of calibration given by Equation (13), a system is perfectly calibrated with respect to $P_r$ if its ouput $\mathbf{q}$ satisfies $q_i = P_r(H_i \mid \mathbf{q})$. This means that, for a perfectly calibrated system, we can simply plug in $q_i$ in place of the posterior in Equation (22) to get optimal decisions. In other words, Bayes decisions made with posteriors provided by a perfectly calibrated system are the best possible decisions for the given system, as long as both calibration and optimality of the decisions are judged with respect to the same distribution, $P_r$. No other strategy for making decisions *based on the system's output* would result in a better expected cost. On the other hand, if we had access to the system's input features, there could very well exist a better decision strategy. A calibrated system does not guarantee that the Bayes decisions are the best that can be made with our system's input, $x$. Rather, it guarantees that they are the best that can be made with our system's *output*, $\mathbf{q}$. If the system is poor, the decisions will be poor, regardless of how well calibrated it may be.

### 2.4.2 Calibration transformations

The posterior $P_r(H_i \mid \mathbf{q})$ in Equation (13), which we have called $s_i$ (Equation 20), can be interpreted as a transformation of the classifier's posterior $\mathbf{q} = P_c(\cdot \mid x)$, into a posterior that is perfectly calibrated with respect to $P_r$. To see this, we can use the proof in Equation (15), substituting $\mathbf{p} \to \mathbf{s}$ and $x \to \mathbf{q}$, to find that $P_r(H_i \mid \mathbf{s}) = s_i$. That is, $\mathbf{s}$ is perfectly calibrated. For this reason, $\mathbf{s} = P_r(. \mid \mathbf{q})$ is called a *calibration transformation*. This transformation needs to be learned from data.

For binary classification tasks, the calibration transformation can be constructed by collecting pairs $\mathbf{q}, h$ for several samples from the task. Noting that $\mathbf{q} = [q_1, 1 - q_1]$, we can quantize $q_1$ into, say, 10 bins and compute the frequency of each class $h$ on each bin to obtain $P_r(h \mid \mathbf{q})$. As we will see, this is the basis for the computation of the ECE metric. Quantizing and counting, though, is often problematic, as we will discuss later in this work, and is not appropriate for multi-class problems. Fortunately, a large variety of calibration transformations that do not rely on quantization have been proposed in the literature over the last decades. A review of these approaches can be found, for example, in the work by Filho et al. (2023).

One of the most standard calibration approaches consists of applying an affine transformation to the logarithm of the posterior vector, training the parameters of this transformation to minimize the cross-entropy. Since the cross-entropy is a strict EPSR, minimizing this loss guides the parameters of the affine transformation toward values that produce good posteriors. Instances of this approach are linear logistic regression, also known as Platt scaling for binary classification (Platt, 2000), an extension of Platt scaling for the multi-class case called direction-preserving (DP) transformation (Brümmer & van Leeuwen, 2006), and temperature scaling (Guo et al., 2017). In our experiments, we will use the DP transformation where the transformed posterior $\mathbf{s} = R(\mathbf{q})$ is given by

$$\mathbf{s} = \text{softmax}(\alpha \log(\mathbf{q}) + \boldsymbol{\beta}) \tag{23}$$

where $\mathbf{q}$, $\mathbf{s}$, and $\boldsymbol{\beta}$ are vectors of dimension $K$, the number of classes, and $\alpha > 0$ is a scalar. When taking $\boldsymbol{\beta} = \mathbf{0}$, this transformation reduces to temperature scaling.[7]

For the binary classification case, it is possible to obtain a monotonic transformation of one of the two posterior probabilities output by the system that leads to the lowest EPSR value on the test data itself. This transformation can be obtained with the pool-adjacent-violator (PAV) algorithm (Ayer et al., 1955). The EPSR resulting after this transformation can be seen as the minimum EPSR possible on that test dataset that does not change the ranking of the posteriors. Interestingly, the transformation is simultaneously optimal for all EPSRs regardless of which one is used to obtain it (Ahuja & Orlin, 1998; Brümmer, 2010). This minimum is likely not achievable on any other dataset, though, since the transform is non-parametric and overfits easily.

To train any calibration transform, a dataset of $\mathbf{q}$ values obtained by running the classifier on a set of samples needs to be created. For supervised approaches like those mentioned above, the class, $h$, for each sample is also required. It is important to note that these samples should not be extracted from the dataset used to train the classifier we are aiming to calibrate. The posteriors that the classifiers produce on their training samples are, for most modern models, already well-calibrated. Yet, unless no degree of overfitting occurred during training, the distribution of those posteriors is not representative of the distribution on unseen data, which is where we wish the calibration transform to perform well. Hence, the data used to train the classifier should not be used to train the calibration model. Ideally, a fraction of the available training data should be held-out when training the classifier to be used as training data for the calibration model. It is particularly important that this data is representative of the one we expect to see when the system is deployed since calibration performance appears to be more sensitive to domain mismatch than discrimination performance (Ferrer et al., 2021).

### 2.5 Calibration metrics

Equation (13) gives the definition of perfect calibration with respect to a reference distribution. It does not provide a calibration metric, that is, a way to measure a degree of (imperfect) calibration. In this section we describe three calibration metrics: the expected divergence, commonly studied in theoretical papers; the expected calibration error, commonly used in empirical machine learning papers; and the calibration loss, our proposed metric, which we believe has various advantages and no disadvantages over both of those well-established metrics.

### 2.5.1 Expected score divergence

Much of the literature on calibration solves the problem of measuring the degree of imperfect calibration by defining a divergence between the left-hand side and the right-hand side in Equation (13), where the left-hand side is given by the output of the system under evaluation, $\mathbf{q}$, and the right-hand side is given by the reference posterior, $\mathbf{s}$. A permissive definition of a divergence is a non-negative function $d(\mathbf{s}, \mathbf{q})$ that is zero if $\mathbf{s} = \mathbf{q}$. A *strict* divergence is zero only at $\mathbf{s} = \mathbf{q}$ and positive, or infinite otherwise. Then, a class of metrics that evaluate the goodness of the calibration of a probabilistic classifier $P_c$, relative to a reference

---

[7]Temperature scaling, as defined by Guo et al. (2017), takes the pre-softmax activations of a neural network as input, instead of the log posterior as in Equation (23). Yet, in both cases the final calibrated output is transformed by the softmax function, making both expressions equivalent.

model $P_r$, can be defined as the expectation of the divergence with respect to the reference distribution (Bröcker, 2009):

$$D(P_r, P_c) = \mathbb{E}_{P_r(\mathbf{q})}\left[d(R(\mathbf{q}); \mathbf{q})\right]. \tag{24}$$

where we defined $R(\mathbf{q}) = \mathbf{s}$, making explicit the fact that $\mathbf{s}$ is a function of $\mathbf{q}$.

Interestingly, it can be shown that EPSRs (Equation 5) can be decomposed into an expected divergence and a generalized entropy term (DeGroot & Fienberg, 1983; Bröcker, 2009; Brümmer, 2010; Gneiting & Raftery, 2007; Dawid, 2007):

$$\mathbb{E}_{P_r(h,\mathbf{q})}\left[C^*(h, \mathbf{q})\right] = \mathbb{E}_{P_r(\mathbf{q})}\left[d(R(\mathbf{q}); \mathbf{q})\right] + \mathbb{E}_{P_r(h,\mathbf{s})}\left[C^*(h, \mathbf{s})\right] \tag{25}$$

where

$$d(\mathbf{s}, \mathbf{q}) = \mathbb{E}_{h\sim\mathbf{s}}\left[C^*(h, \mathbf{q}) - C^*(h, \mathbf{s})\right]. \tag{26}$$

which satisfies the conditions listed above. That is, it is zero if $\mathbf{s} = \mathbf{q}$, and non-negative otherwise. This last property derives directly from the defining property of PSRs, Equation (3). When the PSR is strict, the divergence is zero if and only if $\mathbf{s} = \mathbf{q}$. A divergence that is induced by a PSR as in Equation (26) is called a *score divergence* (Ovcharov, 2015). The score divergence corresponding to the Brier loss is the squared Euclidean distance, $\sum_i (s_i - q_i)^2$, and the one corresponding to the logarithmic loss is the Kullback-Leibler divergence, $\sum_i s_i \log(s_i/q_i)$.

The right-most term in Equation (25) is the EPSR of $\mathbf{s}$ which, for the logarithmic loss, is the entropy of this distribution. This term measures the discrimination or refinement of $\mathbf{q}$: it is the EPSR that is obtained after solving (as best as we could) any miscalibration problem by adding the calibration transform $\mathbf{s}$ to the system. Hence, Equation (25) expresses the overall performance of the system, given by the EPSR on the left-hand side, as a sum of a calibration and a discrimination term.

Note that all terms in the decomposition depend on $P_r$, a model that needs to be learned from data. Different models will lead to different values for all three terms. While this may be unsettling, it is an issue inherent to the fact that the true distribution is not known in practice and, hence, it applies to all metrics designed to assess the performance of posterior probabilities.

Further, note that, in order for Equation (25) to hold, the $P_r(h, \mathbf{q})$ used to obtain the expectation in the left-hand side should be consistent with $\mathbf{s}$. That is, it should satisfy $P_r(H_i, \mathbf{q}) = P_r(\mathbf{q})s_i$. Hence, the left-hand side, which corresponds to the overall performance of the classifier, depends on the output of a calibration model, $\mathbf{s}$. If this model is poor, the assessment of the system performance may be suboptimal (examples of this problem are given in Section 3.4). This makes this classic decomposition from the theoretical literature on calibration particularly problematic in practice.

In contrast, the common practice in the machine learning literature is to compute the EPSR in the left-hand side of Equation (25) by setting $P_r(h, \mathbf{q})$ to be the empirical distribution on the test data, so that the EPSR is given by the average PSR over the data. Unfortunately, doing this in Equation (25) leads to a meaningless decomposition. The posterior $\mathbf{s}$ derived from the empirical distribution consists of a one-hot vector with a one at the index corresponding to the sample's true class for the $\mathbf{q}$ values obtained on the dataset and undefined values for any other value of $\mathbf{q}$. This $\mathbf{s}$ will, for most systems, have an entropy term of 0 (unless two samples from different classes have exactly the same $\mathbf{q}$ value) suggesting that the system has perfect discrimination. Yet, such $\mathbf{s}$ would certainly not result in zero entropy on any other dataset.

In order for the decomposition in Equation (25) to be meaningful, $\mathbf{s}$ needs to be a good predictive model. That is, a model that generalizes well to unseen data. The empirical distribution does not satisfy this condition and, hence, cannot be used for the decomposition. Perhaps for this reason, the divergence/entropy decomposition of EPSRs is not commonly used in empirical papers. As we will see in Section 2.5.3, our proposed calibration loss metric solves this problem.

### 2.5.2 ECE: Expected calibration error

The most common calibration metric in current machine learning literature is the expected calibration error (ECE) (for example, Guo et al., 2017; Liang et al., 2023; Müller et al., 2019; Ovadia et al., 2019), which,

as we will see, is a variant of the expected divergence described above. This metric was proposed by Naeini et al. (2015a) as a metric to assess the calibration quality of binary classification systems. To compute the ECE, the posteriors for class $H_2$ for all the samples in the test dataset are binned into $M$ bins. The ECE is then computed as (Guo et al., 2017):

$$\text{ECE} = \sum_{m=1}^{M} \frac{|B_m|}{N} \left| \text{class2}(B_m) - \text{avep2}(B_m) \right| \tag{27}$$

where $B_m$ is the set of samples for which the posterior for class $H_2$ is in the $m$th bin, $|B_m|$ is the number of samples in that bin, $\text{class2}(B_m)$ is the fraction of samples of class $H_2$ within that bin, and $\text{avep2}(B_m)$ is the average posterior for class $H_2$ for the samples in that bin.

The expression above can be rewritten to look like the divergence in Equation (24), with the expectation taken with respect to the empirical distribution of $\mathbf{q}$, by replacing $|B_m|$ with a sum over all samples in that bin:

$$\text{ECE} \quad = \quad \frac{1}{N} \sum_{t=1}^{N} d(\mathbf{s}(x_t), \hat{\mathbf{q}}(x_t)) \tag{28}$$

where $\hat{\mathbf{q}}$ is such that $\hat{q}_2(x_t) = \text{avep2}(B_{m_t})$, and $\mathbf{s}$ is such that $s_2(x_t) = \text{class2}(B_{m_t})$, where $B_{m_t}$ is the bin corresponding to $q_2(x_t)$, the output of the system for class $H_2$. The $\mathbf{s}$ defined this way corresponds to a very common calibration method called histogram binning (Zadrozny & Elkan, 2001; Naeini et al., 2015b). To obtain the ECE defined by Equation (27), the distance $d$ should be defined as $d(\mathbf{s}, \mathbf{q}) = |s_2 - q_2|$. Unfortunately, the absolute distance is not a score divergence, that is, it is not induced by any PSR. To see this, we can use the fact that all score divergences are Bregman divergences and Bregman divergences satisfy the following property (Ovcharov, 2015):

$$\mathbb{E}[\mathbf{s}] = \arg\min_{\mathbf{q}_0} \mathbb{E}[d(\mathbf{s}; \mathbf{q}_0)] \tag{29}$$

where $\mathbf{q}_0$ is a fixed vector. That is, the divergence between $\mathbf{s}$ and a fixed posterior vector is minimized when that vector is equal to the expectation of $\mathbf{s}$. It is easy to find counterexamples (see Appendix B) where this property is not satisfied for the absolute distance and, hence, we can conclude that the absolute distance is not induced by any PSR. As a consequence, the standard form of the ECE cannot be decomposed as in Equation (25).

The histogram binning calibration transformation used to compute the ECE is, in all works we found that used this metric, always trained on the test data itself. If this transformation overfits the data, the ECE will be overestimated. Alternative versions for the ECE have been proposed where the bins are adapted to contain equal number of samples, showing that the resulting ECE is less prone to overfitting (Nixon et al., 2019). Yet, this only mitigates the problem, without fully solving it. A direct solution to the problem is to treat the calibration transform like any other stage of the system, estimating its parameters on held-out data or through cross-validation on the test data. This, as far as we know, has never been done in the literature where ECE is computed. Further, histogram binning is not necessarily a good calibration transform for every problem (we will show examples of this in the experimental section). If the calibration transform used to compute the calibrated posteriors is not a good match for the problem, then the calibration error will be underestimated since the calibrated posteriors will be poorer than they could be. Again, the ECE could be computed using a different calibration transformation, but this it not what is done in the literature where the ECE definition is always tied to using histogram binning as transform.

Finally, perhaps the biggest weakness of the ECE appears when it is used for multi-class problems. In that case, in order to be able to continue using histogram binning as calibration transform, the multi-class problem is mapped to a new binary problem where only the quality of the confidences is evaluated, ignoring all other values in the posterior vector (Nixon et al., 2019). The confidence is the posterior corresponding to the class selected by the system which, in the literature that uses ECE is taken to be the one with the maximum posterior. This effectively maps the multi-class problem into a binary classification problem: deciding whether the system was correct in its decision by using its confidence as the posterior for correctness.

Given this new problem, one can now use the definition for the ECE, where class2 is the fraction of samples correctly classified by the system and avep2 is the average confidence. We call the multi-class version of ECE, ECEmc. As a consequence of the fact that ECEmc only evaluates confidences rather than the full posterior, it may fail to diagnose calibration problems on any of the other components of the posterior distribution (see Section 3.3). Whenever good posteriors are required, the full vector of posteriors should be evaluated and not just its maximum value (Popordanoska et al., 2022). Notably, in some works, including (Guo et al., 2017), the multi-class definition of ECE is unnecesarily used for binary problems.

While the expression for the ECE in Equation (27) is the one used in most machine learning empirical papers, a more general form for the ECE is sometimes used in theoretical papers where the quantization of the system's output is not done, and the divergence is a general function that satisfies $d(\mathbf{s}, \mathbf{q}) = 0$ when $\mathbf{s} = \mathbf{q}$ and positive otherwise (Widmann et al., 2019). Setting $d$ to be the square euclidean distance, the ECE directly corresponds to the expected divergence for the Brier loss, when $\mathbf{s}$ is obtained by histogram binning. In fact, the transform could also be changed to a general transform in which case the ECE simply coincides with the expected divergence in Equation (24). As we discussed above, though, the expected divergence corresponds to a decomposition of the raw EPSR where the expectation is taken with respect to a distribution based on a calibration model, making it a rather impractical decomposition. The calibration loss proposed in the next section aims to solve this problem.

### 2.5.3 Calibration loss

As discussed earlier in this work, in our view, the main goal of calibration metrics should be to diagnose calibration problems during system development. We then propose the following procedure for computing a calibration metric for that purpose: 1) add a calibration transform at the output of the system, and 2) assess the level of improvement obtained from this additional stage. The level of improvement, measured as the difference in EPSR before and after adding the calibration stage, indicates the degree of miscalibration of the system. We call this metric *calibration loss*. If the calibration loss is large relative to the original EPSR value, then it means the system was poorly calibrated and would benefit from the addition of the proposed calibration stage. If the improvement is relatively small, then it means that the system is already well calibrated. Alternatively, a small relative improvement may be due to the calibration transform being poor. This is the same problem discussed several times above. In order to measure calibration, a reference distribution needs to be created first. The quality of the calibration metric will strongly depend on the quality of this reference distribution. No calibration metric is immune to this issue.

Note that the process to obtain the calibration loss is, essentially, what is done at every step during system development: a new approach is tried (in this case, the addition of a calibration stage) and its benefit is assessed by measuring the relative improvement obtained compared to the baseline (in this case, the system without the calibration stage). As for every other development process, the assessment of the gains will be more reliable if the data where the performance is computed is not used for training or tuning purposes. When computing calibration loss, this means that the test data should not be used to train the calibration transform. This approach which, as far as we know, was never proposed in the calibration literature, ensures that the estimation of the calibration error will not be biased by overfitting effects.

Formally, we define the calibration loss (CalLoss for short) as:

$$\text{CalLoss} = \text{EPSR}_{\text{raw}} - \text{EPSR}_{\text{cal}} \tag{30}$$

where $\text{EPSR}_{\text{raw}}$ and $\text{EPSR}_{\text{cal}}$ are the two EPSRs in Equation (25) with the expectation taken with respect to the empirical distribution, that is, computed as averages over the data, as in Equation (6):

$$\text{EPSR}_{\text{raw}} = \frac{1}{N} \sum_{t=1}^{N} C^*(h_t, \mathbf{q}_t) \tag{31}$$

$$\text{EPSR}_{\text{cal}} = \frac{1}{N} \sum_{t=1}^{N} C^*(h_t, \mathbf{s}_t) \tag{32}$$

These two EPSRs measure the overall performance of the posteriors without and with the addition of the calibration stage, respectively (see Figure 3). As we discussed with regards to Equation (25), $\text{EPSR}_{\text{cal}}$ can

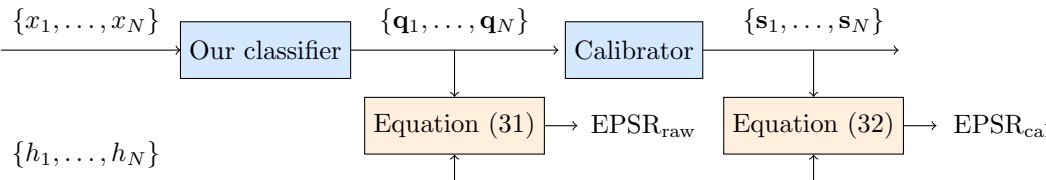

Figure 3: Process for computing the two EPSRs needed to obtain the relative calibration loss metric (Equation 33) using a dataset with $N$ samples.

be seen to measure the inherent discrimination performance of the system since it is the EPSR that remains after the miscalibration has been fixed. Again, this holds as long as the calibration transform is doing a good job at reducing the miscalibration.

For ease of interpretation, in our experiments we report relative CalLoss, RCL for short:

$$\text{RCL} = 100 \; (\text{EPSR}_{\text{raw}} - \text{EPSR}_{\text{cal}}) / \text{EPSR}_{\text{raw}} \,. \tag{33}$$

The RCL reflects the relative improvement in EPSR that we would be able to achieve by adding a post-hoc calibration stage given by $\mathbf{s}$ to our system.

CalLoss differs from the divergence defined by Equation (24) in the way the expectations of the PSRs are computed on both sides. Instead of computing them with respect to a distribution consistent with $\mathbf{s}$ they are computed with respect to the empirical distribution. As a consequence, the calibration term is no longer an expected score divergence. Instead, it is simply the difference between the two empirical EPSRs. As we will show in the experimental section, as long as $\mathbf{s}$ is a good calibration model, the EPSRs computed as in Equation (24) or as in Equation (30) are very similar, resulting in the same value for the calibration metric, which is the difference between the two EPSRs. On the other hand, when $\mathbf{s}$ is not a good calibration model, the left-hand side in Equation (25) can be a very poor estimator of the classifier's performance, resulting, as a consequence, in a meaningless decomposition. In contrast, $\text{EPSR}_{\text{raw}}$ computed as in Equation (31) does not depend on a calibration model, resulting in a more reliable measure of the overall performance of the classifier. Even if the calibration model is poor, the calibration loss value is still meaningful: it indicates the improvement in EPSR that can be achieved by using that same calibration model as part of the system.

A specific instance of the calibration loss metric was proposed almost two decades ago for the binary classification problem of speaker verification by Brümmer & du Preez (2006). This metric which, as far as we know, has since only been used for the speaker verification and forensic applications (Ramos et al., 2017; 2020), is a special case of our proposed approach above for binary classification where the calibration transform is given by the best isotonic transformation obtained on the test data itself. In this work, we consider a general form of the calibration loss where the transform can be adapted to the problem of interest and, preferably, trained on held-out data rather than on the test data itself. This generalization allows us to use the metric for multi-class problems where the isotonic transformation is not applicable and to avoid the bias produced by the train-on-test approach.

## 3 Experiments

In this section we show an analysis of the metrics discussed in the previous section using both synthetic and real datasets. Synthetic datasets allow us to have the ground-truth distribution, which in turn allows us to obtain perfectly-calibrated posteriors where we would expect any good calibration metric to give a value of zero. As we will see, this does not always happen, allowing us to revisit some of the theoretical issues discussed in previous sections.

For synthetic data, the procedure for generating calibrated and miscalibrated posteriors is described in Appendix C. Briefly, $N$ samples are generated by sampling the input features from a multivariate Gaussian distribution for each class. The number of samples generated for each class is determined by an imbalanced prior distribution with a prior for class $H_1$ of 0.8 and equal prior for other classes, unless otherwise noted. The per-class multivariate Gaussian distributions are determined such that the means are equidistant at an L2 distance of 1. The covariance is diagonal with same variance for all dimensions and shared across all classes and it controls the performance of the resulting posteriors. We set the variance to 0.15 unless otherwise indicated. Perfectly calibrated posteriors are obtained using the known feature distribution for each class. This is the optimal classifier for this data since it is derived from the generating distribution. Then, three miscalibrated versions of these posteriors are created, one by artificially manipulating the prior distribution to be mismatched to that in the data (mcp), another one by scaling the logarithm of the posteriors and renormalizing the result to produce overconfident or underconfident posteriors depending on the scale (mcs), and a final one combining both causes of miscalibration (mcps).

In Section 3.1 through 3.4 we show and discuss results for datasets with 2 and 10 classes, including first an illustration of the property described in Section 2.3.4 and then focusing on the comparison of various EPSRs and calibration metrics. Finally, in Section 3.5, results on real datasets are presented and discussed, showing the effect of post-hoc calibration on the different metrics on posteriors produced by actual systems.

In practice, it is important to assess the robustness of the performance estimate obtained on our test dataset, specially when the number of samples is small. Appendix E includes an example on how to obtain confidence intervals using bootstrapping. We do not include confidence intervals in the results in this section to keep both the code that produces them and the plots simple.

### 3.1 Illustration of EPSR construction as integrals over Bayes risks

In this section we illustrate the construction of CE and BS as integrals over Bayes risks (see Section 2.3.4) using synthetic posteriors for a 2-class dataset. We generate posteriors for four different classifiers as described above and further detailed in Appendix C:

- **cal**: Perfectly calibrated system obtained with a per-class variance of 0.15.

- **mcs-u**: Miscalibrated system obtained by scaling the log posteriors from the cal system by 0.48 to simulate an underconfident system, and then converting the result back into posteriors by doing softmax.

- **mcs-o**: Same as mcs-u but scaling the log posteriors by 2.0, to simulate an overconfident system.

- **cal-h**: Same the cal system but with a variance of 0.19 to obtain a harder dataset.

In all cases, the prior for class $H_1$ is set to 0.6 and the total number of samples is set to 100,000.

The left plot in Figure 4 shows the Bayes risk for those four systems for cost matrices given by

$$C_{\mathbf{a}} = \begin{bmatrix} 0 & 1/a_1 \\ 1/(1-a_1) & 0 \end{bmatrix} \tag{34}$$

while varying the value of $a_1$ between 0 and 1. The CE for each system is simply the area under the corresponding curve. Systems cal, mcs-u, and mcs-o have the same Bayes risk for $a_1 = 0.5$, which corresponds to the total error rate multiplied by 2. The corresponding Bayes decisions are given by the argmax rule. Since mcs-u and mcs-o are constructed by scaling the log posteriors of the cal system, which does not affect the argmax decisions, the total error rate is the same for all three systems. Yet, since the Bayes risk at other values of $a_1$ differs across systems, the CE also differs, being higher for the two miscalibrated systems. Finally, we can see that the calibrated posteriors for the harder dataset, cal-h, have a worse total error rate than the other three systems, but better Bayes risk than the mcs posteriors at extreme values of $a_1$. Overall, integrating the curves for mcs-u, mcs-o and cal-h leads to the same NCE values (recall that NCE $= \mathrm{CE}/(-\sum P_i \log P_i)$, with $P_1 = 0.6$ and $P_2 = 0.4$ in these synthetic datasets).

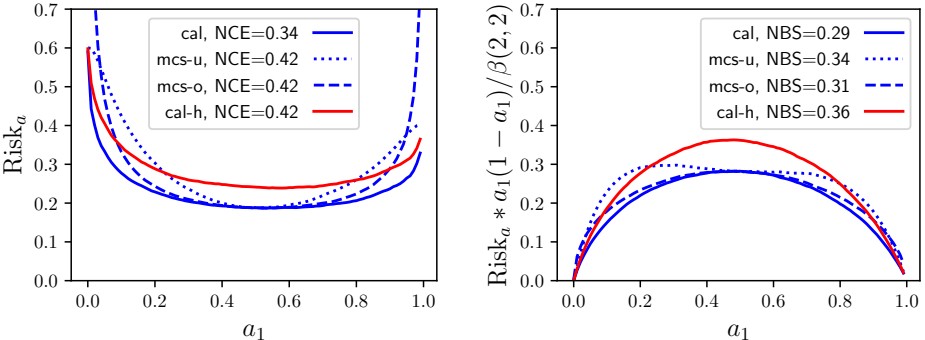

Figure 4: Weighted Bayes risk curves for four different systems. Left: the weight is 1.0 so that the integral under these curves is the CE. Right: the weight is $a_1 (1 - a_1)/\beta(2, 2)$ so that the integral is the BS. The normalized CE, NCE, and normalized BS, NBS, are shown in the legends.

The right plot in Figure 4 shows the Bayes risk multiplied by $W(\mathbf{s}) = a_1 (1 - a_1)/\beta(2, 2)$ so that the area under these curves corresponds to the BS. We can see that the BS de-emphasizes the performance of the system at extreme values of $a_1$. Note that the Bayes decisions corresponding to the cost matrix defined by Equation (34) are given by thresholding the posterior for class $H_1$ at a threshold of $a_1$ (this can be derived by replacing $C(H_1, D_2) = 1/a_1$ and $C(H_2, D_1) = 1/(1 - a_1)$ in Equation (1)). Hence, the operating points that are deemphasized by BS are those corresponding to extreme threshold values. Poor posteriors in those regions will be penalized much less by BS than by CE. A similar conclusions can be derived from comparing the log-loss expression corresponding to the CE and the squared loss expression corresponding to BS (see Figure 1). While the squared loss is bounded by one, which happens when the posterior for the true class is zero, the log-loss gives infinite penalization to those errors. This difference between CE and the BS results in a difference in ranking of the four systems according to each metric. While systems mcs-u, mcs-o, and cal-h are identical in terms of CE, they are not so in terms of BS due to the different weighting given to each operating point.

As is clear from this example, different EPSRs may result in different development decisions. Hence, it is important to choose an EPSR that correctly reflects the needs of the application. Expression (10) provides an intuitive way of creating new PSRs, allowing us to select a weight function $W$ appropriate for the task. By default, though, if no specific needs are identified, the uniform weighting function, which results in the CE metric, is a good general choice. The BS may not be a desirable choice for high-stakes application due to the effect observed in Figure 4 where the behavior of the risk for extremely imbalanced cost matrices is deemphasized. This means that if a system produces poor posteriors in the extremes (very high or very low values), the BS will not correctly diagnose the problem. The CE, on the other hand, will severely penalize such systems, making it a more appropriate choice for high-stakes applications where having good extreme posteriors is important.

The plots in Figure 4 also help us illustrate our argument about the role that calibration should (or, rather, should not) play in the evaluation of posteriors. While system mcs-o is miscalibrated, it is better than system cal-h which is perfectly calibrated in terms of BS (and the same in terms of CE). If we had decided that BS is the EPSR of choice for our problem, there would be no reason to select the cal-h system over the mcs-o system, despite the fact that the latter system is miscalibrated. If we had the chance to add a post-hoc calibration stage to our system, we could compute the calibration loss to see that the mcs-o posteriors are miscalibrated which would indicate that a calibration stage is needed. Yet, after adding that stage, the performance of the system should again be assessed in terms of its new EPSR.

## 3.2 Results on a Binary classification problem

The left plot in Figure 5 shows five different metrics for the 2-class synthetic dataset, using the procedure described in Appendix C to produce 400 samples. We chose to use a relatively small dataset to show the effect that overfitting of the calibration transform may have in some of the metrics. The metrics include

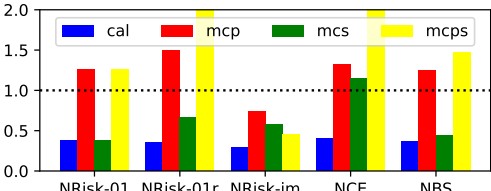 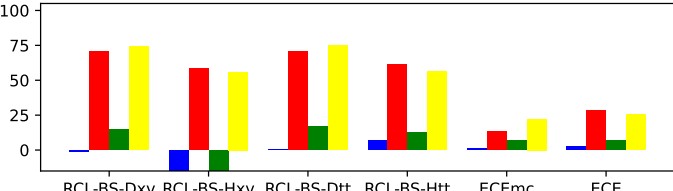

Figure 5: Various metrics for a binary classification task for four different systems, one perfectly calibrated (cal) and three miscalibrated ones (mcs, mcp, mcps). Left: normalized overall performance metrics. The dashed line indicates the performance of a naive system. Right: calibration metrics including the binary ECE, the multiclass ECE (ECEmc), and the RCL based on BS for two calibration approaches, DP (D), and histogram binning (H), trained either through cross-validation on the test data (xv) or training on the full test dataset (tt).

normalized cross-entropy (NCE), normalized Brier score (NBS), and three normalized Bayes risks, with cost matrices given by 1) the 0-1 cost matrix (NRisk-01), 2) the 0-1 cost matrix with an additional column corresponding to a reject decision with cost of 0.1 for both classes (NRisk-01r), and 3) a square imbalanced cost matrix with $c_{12} = 1$ and $c_{21} = 10$ (NRisk-imb). The accuracy for each system can be computed as $1 - 0.2$ NRisk-01, since NRisk-01 = Risk-01/0.2, where Risk-01 is total error rate.

The results show that NRisk-01 is unaffected by the miscalibration in mcs, a scaling in the logarithm of the posteriors which results in overconfident posteriors. This happens because the argmax decisions, which are the Bayes decisions for this cost function, are not affected by the scaling. NRisk-01r and NRisk-im, on the other hand, are affected by this type of miscalibration, with NRisk-im being less affected than NRisk-01 by the mismatch in priors in mcp. As expected, we see that the risks have different behavior depending on the cost matrix, highlighting the importance of correctly selecting this matrix for the problem of interest when categorical decisions are required.

The three risks in the figure are EPSRs since they are computed using Bayes decisions. Yet, they are not strict PSRs. They only assess the performance of the posteriors for one specific operating point given by the cost matrix, which corresponds to a set of decision regions determined by Equation (1). The decisions regions are, in turn, used to make categorical decisions which are then used to compute the risk. Hence, two systems that results on the same categorical decisions for a given cost matrix would have the same risk value, while one may have better posteriors than the other for other cost matrices.

If we want to assess the goodness of the posteriors in general, across the full simplex, we need to use the expectation of strict PSRs, like the CE or BS. These two metrics assess the goodness of the posteriors without going through the step of making categorical decisions. As discussed in Section 3.1, despite both metrics being strict PSRs, they may result in different conclusions about the quality of the posteriors. This can be observed in the left plot in Figure 5 where the normalized CE and BS metrics show a rather different assessment of the quality of the mcs and mcps systems. This happens because the overconfident posteriors from the mcs and mcps systems are pushed to the extremes, where the BS pays little attention.

The right plot in Figure 5 shows various calibration metrics: binary ECE, multiclass ECE, and various RCL metrics. The RCL metrics are computed using histogram binning or DP calibration trained with cross-validation or on the test data to obtain $\text{EPSR}_{\text{cal}}$ (Equation 32). We can see that the histogram binning approach is problematic for the cal and mcs datasets. When using cross-validation to train it (as for RCL-BS-Hxv), the transform overfits the training data for each fold resulting in bad calibration when applying that transform on the test data for that fold. Hence, the posteriors are actually worse than the original ones, resulting in a negative calibration loss, something that should never happen for a well-designed well-trained calibration transform. When training on the test data (as for RCL-BS-Htt), the transform overfits the data and the metric diagnoses a non-existing calibration problem on the cal posteriors. These results show that, for this particular problem, histogram binning is a poor calibration approach. In contrast, the RCL values obtained with the DP transform are quite robust for both training approaches, since the

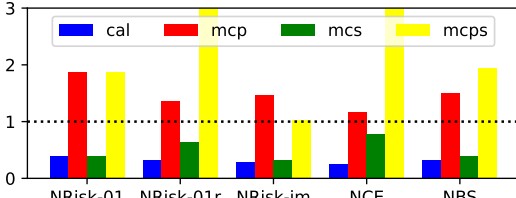 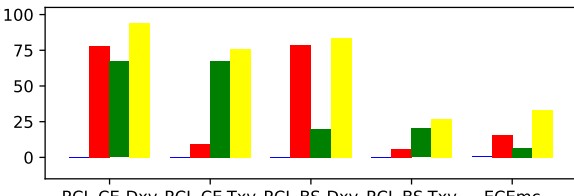

Figure 6: Overall and calibration metrics as in Figure 5 but for a 10-class classfication task.

number of parameters in this transform is small and it is much less prone to overfitting. Both of these RCL metrics correctly diagnose the calibration problem in the mc posteriors, and the correct calibration in the cal posteriors. The value of these metrics tell us the relative improvement in BS that we would get from adding an DP calibration stage to the system and, hence, directly quantify how badly calibrated the original posteriors are without this stage.

Note that, given the way the posteriors are generated in our synthetic datasets, the DP transformation is, by design, able to perfectly reverse the miscalibration present in those posteriors. In other datasets, the DP transformation may not be necessarily optimal since misscalibration can potentially have non-linear effects in the posteriors. Other calibration transformations available in the literature could be explored in those cases. In our experience, though, DP calibration has given excellent results across a variety of tasks. We will see further evidence of this in Section 3.5. As discussed in the next section, this is not the case for temperature scaling, which fails in cases where the prior distribution in the training and the test data are different, a common scenario in many applications.

Turning to the ECE metrics, we can see that ECEmc gives a different estimate of the calibration error than the original binary ECE, which is expected since ECEmc is evaluating the performance of a different binary problem, as explained in Section 2.5.2. We can also see that the ECE shows a similar trend as the RCL-BS-Htt since these two metrics use the same calibrated posteriors obtained with histogram binning trained on the test data, differing only in the way the distance between the raw posteriors and the calibrated posteriors is computed (see Section 2.5.2). Hence, the ECE suffers from the same problem as RCL-BS-Htt discussed above, mistakenly diagnosing a small calibration problem in the cal posteriors. Finally, note that the value of the ECE does not have a clear interpretation, while, as discussed above, the RCL values directly indicate the percentage of the EPSR of the original posteriors which can be reduced by doing calibration, providing a more actionable result.

## 3.3 Results on a Multi-class classification problem

In this section we show results for a multi-class classification task instead of a binary task as above, including additional versions of the calibration loss, using CE as the EPSR as well as BS, and using temperature scaling as well as DP calibration to obtain the calibrated posteriors needed to compute that metric. The dataset was created following the process described in Appendix C with $K = 10$ classes, feature variance of 0.08, and a total number of samples $N = 2000$.

The left plot in Figure 6 shows the results for various overall metrics. The three risks are a direct generalization of the ones for the 2-class case: 1) the 0-1 cost matrix (NRisk-01), 2) the 0-1 cost matrix with an additional column corresponding to a reject decision with costs equal to 0.1 for all classes (NRisk-01r), and 3) a square cost matrix with imbalanced costs, $c_{ij} = 1$ for all $i \neq j$, except when $i = 10$ in which case $c_{ij} = 10$ (NRisk-imb). The conclusion from these results is similar to that for the binary case: each risk gives a different ranking of systems, and the NCE is more severely degraded by the scaling miscalibration in mcs and mcps than the NBS.

The right plot in this figure shows the relative CalLoss (RCL) based on BS and CE for two calibration methods, DP calibration and temperature scaling trained with cross-validation, and the multiclass ECE. These results show a large difference between some of the calibration metrics. Those that use temperature

| System | cal | | | mcs | | | mcp | | | mcps | | |
|---|---|---|---|---|---|---|---|---|---|---|---|---|
| #Samples | 200 | 2000 | 20000 | 200 | 2000 | 20000 | 200 | 2000 | 20000 | 200 | 2000 | 20000 |
| true | -3 | -5 | -2 | -7 | -10 | -5 | 4 | -2 | 0 | 4 | -2 | 0 |
| Dtt | 0 | 0 | 0 | 0 | 0 | 0 | 0 | 0 | 0 | 0 | 0 | 0 |
| Dxv | 1 | 0 | 0 | 1 | 0 | 0 | 1 | 0 | 0 | 0 | 0 | 0 |
| Ttt | 1 | -1 | 0 | 1 | -1 | 0 | 29 | 28 | 28 | 29 | 28 | 28 |
| Txv | 1 | -1 | 0 | 2 | -1 | 0 | 29 | 28 | 28 | 29 | 28 | 28 |
| Htt | -6 | -8 | -9 | -153 | -143 | -132 | -19 | -21 | -17 | -62 | -84 | -82 |
| Hxv | -2 | -8 | -9 | -171 | -144 | -132 | -15 | -21 | -17 | -62 | -83 | -82 |

Table 1: Relative difference between the semi-empirical cross-entropy ($\text{CE}_{se}$) and the empirical cross-entropy ($\text{CE}_e$) computed as $(\text{CE}_e - \text{CE}_{se})/\text{CE}_e * 100$ and then rounded to the nearest integer. Results are shown for synthetic posteriors as in Figure 5, for different number of samples. The semi-empirical cross-entropy is computed with respect to posteriors given by the true posterior (true), and by calibrated versions of the posteriors using DP calibration (D), temperature scaling (T), and histogram binning (H) trained with cross-validation (xv) or trained on the test data (tt).

scaling for calibration fail to diagnose the severity of the calibration problem in mcp. This is because this method cannot compensate for calibration problems due to mismatched priors and, hence, it also cannot be used to diagnose them. As any other calibration metric, CalLoss fails if the calibration transform is not a good match for the problem. Yet, the CalLoss framework allows us to explore a variety of transforms. If one such transform leads to a relatively large CalLoss, and that transform was not trained on the test data itself, then we can conclude that the system is miscalibrated. Importantly, the same transform that was used to diagnose the problem can be used as post-hoc calibration stage and reduce the EPSR by a percentage indicated by the RCL value.

Turning to ECEmc, we can see that it gives a different assessement of the relative severity of the miscalibration of the different datasets compared to the RCL metrics. This is partly because it uses a different way to measure the distance between the raw and calibrated scores—one that is not induced by any PSR—and partly because it only evaluates the quality of the maximum posterior for each sample rather than the full vector. When the miscalibration occurs on classes other than the one with maximum posterior, the ECEmc metric cannot properly diagnose the problem. This issue with the multi-class ECE has also been discussed by Nixon et al. (2019). The CalLoss metric does not suffer from this problem as it assesses the quality of the full posterior vector. In addition, note that the scale of the ECEmc is not interpretable. While, for example, the RCL-CE-Dxv values for miscalibrated systems indicate that over 70% of the CE is due to miscalibration, no equivalent interpretation is possible with the ECEmc.

### 3.4 Effect of the reference distribution in the EPSR value

As explained in Section 2.5, EPSRs values may differ depending on the reference distribution used to take the expectation. In this section, we compare cross-entropy values (the expectation of the NLL) given by:

$$\mathbb{E}_{P_r(h,\mathbf{q})}\left[C^*(h, \mathbf{q})\right] = \mathbb{E}_{P_r(h,\mathbf{q})}\left[-\log(q_h)\right] \tag{35}$$

for different reference distributions: the empirical distribution in the test data used in all prior results in this section (Equation 8) and various *semi-empirical* distributions where $P_r(\mathbf{q})$ is the empirical distribution for $\mathbf{q}$, but $P_r(h \mid \mathbf{q})$ is given by a classifier. Since we are working with synthetic data, we can derive $P_r(h \mid \mathbf{q})$ from the true generating distribution. This posterior, of course, would not be available on real data. Hence, we compare with different $P_r(h \mid \mathbf{q})$ given by calibrating the posteriors using histogram binning, DP calibration, and temperature scaling calibration trained with cross-validation or directly on the test data.

Table 1 shows the relative difference between the semi-empirical cross-entropy and the empirical cross-entropy for the same four datasets used in Figure 5 with varying number of samples. We can see that, in most cases, the relative difference between the empirical and the semi-empirical cross-entropy is very small, specially for the larger datasets where the empirical cross-entropy is less noisy. Larger differences occur when the calibration transformation does not fully fix the calibration problem, which happens with temperature scaling for the mcs and mcps posteriors and for histogram binning in most cases, even for the larger datasets.

These examples illustrate the practical problem involved in using the expected score divergence as a calibration metric. If the calibration model used to obtain $\mathbf{s}(\mathbf{q}) = P_r(. \mid \mathbf{q})$ in Equation (25) does not do a good job at calibrating the posteriors, the EPSR in the left-hand side may be a poor estimate of the system performance. This will, in turn, result in a meaningless decomposition with the two terms adding up to a value that does not reflect the actual performance of the system. On the other hand, CalLoss is meaningful regardless of the quality of the calibration model. The $\text{EPSR}_{\text{raw}}$ and $\text{EPSR}_{\text{cal}}$ used to compute it reflect the performance of the system before and after calibration with that—perhaps suboptimal—calibration model. Hence, CalLoss always predicts the level of improvement in EPSR that can be obtained from using the selected calibration transform. For this reason, we believe that, in practice, CalLoss is a better calibration metric than the expected score divergence.

## 3.5 Results on real datasets

In this section, we show results on real datasets for a number of different tasks corresponding to speech, image, and natural language processing (NLP) tasks.

For NLP we include SST2 and AGNEWS. SST2 (Socher et al., 2013) is a natural language processing dataset where the task is to decide whether a certain text has positive or negative sentiment. In AGNEWS (Gulli, 2005; Zhang et al., 2015) the task is to classify news into 4 different classes. The posteriors for these datasets were produced with the GPT-2 model using the code provided in `https://github.com/LautaroEst/efficient-reestimation` using zero-shot prompts.

For speech processing we include three datasets, SITW, FVCAUS and IEMOCAP. SITW (McLaren et al., 2016) and FVCAUS (Morrison et al., 2015; 2012) are two speaker verification datasets where the task is to decide whether two audio samples belong to the same speaker or not. To obtain posteriors for these two datasets, we ran an X-vector PLDA system using the code provided in `https://github.com/luferrer/DCA-PLDA`. IEMOCAP is a speech processing dataset where the task is to classify each speech sample into a set of emotions: angry, happy, sad, and neutral. The posteriors were downloaded from `https://github.com/habla-liaa/ser-with-w2v2/tree/master/experiments/w2v2PT-fusion`.

Finally, for image processing we include results on CIFAR10, which is an image processing datasets where the task is to classify the object in an image into one of 10 classes. The posteriors for this dataset were obtained using the code available in `https://github.com/chenyaofo/image-classification-codebase`. We evaluate three models: resnet20, vgg19, and repvgg_a2. These models have approximately 0.27 million, 20 million, and 27 million parameters, respectively. We also include three medical imaging tasks from the MedMNIST dataset (Yang et al., 2023): PATH, composed of histological images where the task is to classify into 9 types of tissues; PNEUM, composed of pediatric chest x-rays classified as pneumonia or normal; and ADRENAL, composed of tomography scans of adrenal glands classified as "normal adrenal" or "adrenal mass". The posteriors were downloaded from `https://zenodo.org/records/7782114` for the Resnet-50 models for the maximum image resolution.

We also created binary classification tasks for the detection of one specific class versus all others for CIFAR100, a dataset similar to CIFAR10 but with 100 classes, using the posterior obtained with a resnet20 system (using the same codebase as above for CIFAR10) for that class and 1 minus that posterior for the "other" class. We call these posteriors CIFAR-XvsO, where X identifies the target class.

Table 3 in Appendix D shows the number of classes, the priors, and the total number of samples for all the datasets. SST2, SITW and FVCAUS as well as the CIGAR-XvsO sets are binary classification problems, while all others are multi-class problems.

For each dataset and system, we show results obtained on the raw posteriors as they come out of the system, and on calibrated posteriors using DP calibration trained with cross-validation on the test data. For the binary classification tasks, we also show results for the best calibrated posteriors obtained with the PAV algorithm run on the full test set. Table 2 shows three normalized risk metrics, and NCE, RCL, ECE, and ECEmc for each of those systems.

Focusing on the calibration metrics for the raw posteriors (see entries highlighted in red), we can see that both ECE metrics often fail to diagnose calibration problems. For example, according to ECE, the SITW raw

| Dataset | System | proc | NRisk | | | | Calibration | | |
|---|---|---|---|---|---|---|---|---|---|
| | | | C01 | Cab | Cimb | NCE | RCL | ECEmc | ECE |
| SST2 | GPT2-4sh | raw | 0.996 | 0.812 | 1.000 | 1.073 | 62.4 | 34.4 | 34.8 |
| | | cal | 0.226 | 0.585 | 0.711 | 0.404 | -0.6 | 1.6 | 2.2 |
| | | calp | 0.223 | 0.555 | 0.660 | 0.385 | -0.4 | 0.0 | 0.0 |
| | GPT2-0sh | raw | 0.828 | 0.818 | 1.000 | 0.917 | 46.0 | 20.0 | 27.3 |
| | | cal | 0.310 | 0.655 | 0.685 | 0.495 | -0.6 | 1.4 | 1.5 |
| | | calp | 0.298 | 0.638 | 0.661 | 0.478 | -0.4 | 0.0 | 0.0 |
| SITW | XvPLDA | raw | 0.324 | 0.225 | 0.191 | 0.189 | 16.7 | 0.2 | 0.2 |
| | | cal | 0.306 | 0.190 | 0.153 | 0.158 | -0.1 | 0.0 | 0.0 |
| | | calp | 0.304 | 0.188 | 0.152 | 0.155 | -0.0 | 0.0 | 0.0 |
| FVCAUS | XvPLDA | raw | 3.915 | 1.992 | 1.049 | 1.966 | 99.6 | 1.8 | 8.6 |
| | | cal | 0.012 | 0.008 | 0.005 | 0.008 | -1.0 | 0.0 | 0.0 |
| | | calp | 0.012 | 0.007 | 0.004 | 0.006 | -0.9 | 0.0 | 0.0 |
| CIFAR-1vsO | Resnet-20 | raw | 0.420 | 0.236 | 0.158 | 0.214 | 7.1 | 0.2 | 0.2 |
| | | cal | 0.430 | 0.199 | 0.130 | 0.199 | -2.3 | 0.1 | 0.2 |
| | | calp | 0.400 | 0.175 | 0.132 | 0.169 | -1.1 | 0.0 | 0.0 |
| CIFAR-2vsO | Resnet-20 | raw | 0.700 | 0.442 | 0.371 | 0.393 | 8.2 | 0.3 | 0.4 |
| | | cal | 0.560 | 0.440 | 0.367 | 0.361 | -0.4 | 0.3 | 0.3 |
| | | calp | 0.560 | 0.415 | 0.363 | 0.329 | -0.5 | 0.0 | 0.0 |
| PNEUM | Resnet-50 | raw | 0.278 | 0.736 | 0.479 | 0.802 | 58.1 | 7.7 | 8.6 |
| | | cal | 0.239 | 0.433 | 0.453 | 0.336 | -2.0 | 2.0 | 2.4 |
| | | calp | 0.218 | 0.404 | 0.449 | 0.302 | -1.8 | 0.0 | 0.0 |
| ADRENAL | Resnet-50 | raw | 0.928 | 1.013 | 1.122 | 0.931 | 15.9 | 11.1 | 12.0 |
| | | cal | 0.913 | 0.815 | 0.747 | 0.783 | -1.3 | 3.7 | 4.3 |
| | | calp | 0.826 | 0.705 | 0.668 | 0.714 | -0.9 | 0.0 | 0.0 |
| PATH | Resnet-50 | raw | 0.114 | 0.684 | 0.142 | 0.329 | 69.0 | 7.2 | - |
| | | cal | 0.076 | 0.281 | 0.106 | 0.102 | -0.6 | 1.4 | - |
| IEMOCAP | W2V2 | raw | 0.504 | 1.056 | 0.607 | 0.635 | 3.1 | 6.3 | - |
| | | cal | 0.494 | 0.984 | 0.606 | 0.615 | -0.1 | 2.7 | - |
| AGNEWS | GPT2-0sh | raw | 0.780 | 1.009 | 0.936 | 0.814 | 34.2 | 18.4 | - |
| | | cal | 0.378 | 1.014 | 0.762 | 0.536 | -0.1 | 3.7 | - |
| CIFAR10 | Resnet-20 | raw | 0.082 | 0.406 | 0.121 | 0.122 | 17.2 | 3.9 | - |
| | | cal | 0.083 | 0.311 | 0.112 | 0.101 | -0.8 | 0.8 | - |
| | Vgg19 | raw | 0.068 | 0.486 | 0.100 | 0.153 | 32.4 | 5.0 | - |
| | | cal | 0.069 | 0.322 | 0.097 | 0.103 | -0.6 | 1.7 | - |
| | RepVgg-a2 | raw | 0.053 | 0.309 | 0.076 | 0.092 | 19.9 | 3.2 | - |
| | | cal | 0.052 | 0.239 | 0.067 | 0.074 | -1.2 | 0.9 | - |

Table 2: Various metrics on raw and calibrated posteriors for different speech, image and natural language processing datasets. Calibrated posteriors (cal) are obtained with DP calibration using a cross-validation procedure. For the binary tasks, calibrated posteriors obtained with the PAV algorithm (calp) are also considered. We report three normalized risk values (NRisk) for the same cost matrices as in Figure 5, normalized empirical cross-entropy (NCE), relative calibration loss for CE using DP calibration with cross-validation (RCL-CE-Dxv in Figures 5 and 6, here RCL for short), binary ECE, and multi-class ECE (ECEmc). Colored entries highlight comparisons made in the text.

posteriors are calibrated, while we know they are not since their NCE improves significantly after calibration. Similarly, while for FVCAUS the miscalibration is to blame for almost 100% of the NCE value, ECE is below 10 and ECEmc is close to 0: both ECE metrics strikingly fail to diagnose the severity of the miscalibration of this system. A similar thing happens for PNEUM and PATH, where the ECE values are relatively low, but the RCL is quite large. On the other hand, for ADRENAL, the ECEs are higher than for PNEUM and PATH, but the RCL is much lower. Overall, we can see a very weak relationship between the ECEs and the gain that can be achieved from calibration, given by the difference between the NCE of the raw and the calibrated systems. This indicates that the ECE is not a useful tool for diagnosing misscalibration. On the other hand, the RCL is defined as the relative gain that can be achieved from post-hoc calibration, directly indicating the impact that adding such a stage would have on the system.

In some cases, ECEmc suggests that a miscalibration problem exists where RCL does not. For example, for the CIFAR100 RepVgg-a2 raw posteriors, the ECEmc of 5.6 indicates a small calibration error while the RCL of 0.8 suggest the system is very well-calibrated. First, it is important to note that the ECEmc is not a relative measure. It is not possible to assess the impact that an ECE value of 5.6 would have on the performance of a system. In fact, the impact will be different depending on the overall performance of the system. Unfortunately, this metric cannot be turned into a relative value like the RCL because there is no corresponding NCE to use as reference. In contrast, an RCL value of 0.8 means that only 0.8% of the NCE value is due to miscalibration. Further, note that ECEmc only assesses the performance of the maximum posterior for each sample while RCL assesses the performance of the full vector of posteriors. The ECEmc would tend to overestimate the impact of a small miscalibration in the maximum posterior since it disregards the other posteriors (99 of them in this 100-class task) which may be very well calibrated. Finally, it is possible that the low RCL is due to the DP calibration transformation not being enough to fix the misscalibration problem, resulting in an underestimation of the RCL. In practice, a developer could explore several calibration alternatives before deciding on one approach. The best calibration approach is the one that results in the lowest EPSR value (and, hence, the largest RCL), as long as the calibration model was not trained on the test data. Exploration of various calibration techniques is out of the scope of this paper.

Another important observation from this table is the fact that, sometimes, a miscalibrated system is better in terms of the quality of its posteriors than one that is well calibrated. See, for example, the results for the CIFAR10 posteriors (entries highlighted in green). The Resnet-20 Dxv-calibrated posteriors have perfect calibration and an NCE of 0.101. The RepVgg-a2 raw posteriors, on the other hand, are miscalibrated with an RCL of 19.9%, but have an NCE of 0.092. This means that the posterior probabilities generated by the latter system are better. Further evidence of this can be found in the NRisk values, which are consistently better for the RepVgg-a2 raw posteriors than for the Resnet-20 calibrates ones. This example illustrates why assessing calibration is not helpful to determine the quality of posterior probabilities and should never be used to compare systems with each other.

Results for CIFAR10 also illustrate the fact that two systems with the same NCE are not necessarily equally good for every operating point, as determined by a specific risk function. For example, the Resnet-20 and Vgg19 Dxv-calibrated posteriors have almost identical NCE (0.101 vs 0.103), but show different trends for the three NRisk metrics (see entries highlighted in blue and the green ones above). When the application of interest has a specific well-defined risk function, then that metric should be used to make development decisions rather than NCE, which integrates over all operating points instead of focusing on the one relevant for the task.

Comparing the results for Dxv-calibrated (cal) and PAV-calibrated posteriors (calp) we can see that they are very similar in most cases (see entries highlighted in orange). This implies that the Dxv-calibration procedure is doing a near optimal job at calibrating the posteriors for those datasets. The small difference between cal and calp may be due to a number of reasons: 1) that the DP transformation is not sufficiently expressive for these datasets, 2) that the transforms trained with cross-validation are not generalizing well to the test sets due to the small number of samples available, or 3) that the isotonic PAV transform overfitted the test data resulting in an unrealistically low NCE value. The only way to determine whether the NCE for calp posteriors is too low or the one for cal posteriors is too large is to keep working on the calibration transformation in search of one that, when trained on held-out data or through cross-validation, reduces the NCE further than Dxv calibration.

## 4 Discussion and conclusions

In this work, we focus on the problem of evaluating the quality of probabilistic classifiers. We review the classical concept of proper scoring rules (PSRs) which are scoring functions designed to assess the performance of probability distributions. The expectation of a PSR (EPSR) reflects the quality of the posterior probabilities provided by a classifier. When comparing two systems, a lower EPSR indicates that the system's posteriors are better for making decisions and, hence, also better for interpretation by an end user. A normalized EPSR larger than 1.0 indicates that making decisions with those posteriors would be worse than making decisions simply based on the class priors.

Despite the existence of this elegant and principled tool to assess the performance of probabilistic classifiers, the current trend in much of the machine learning literature is to use calibration metrics for this purpose, under the claim that good calibration is an essential requirement for posteriors to be interpretable, safe, and reliable. In this work, we contend this view and argue that good calibration is neither necessary nor sufficient for a probabilistic classifier's posterior to be of value to the end user. A system can have a relatively high calibration error and still be useful to the user, as long as its EPSR is low. On the other hand, a system might be perfectly calibrated but provide very little value to the user if its EPSR is too high. Hence, we see no reason to report calibration metrics as a way to assess the value of a probabilistic classifier's outputs.

It is important to note, though, that there is also usually no reason to tolerate a miscalibrated system. Any miscalibrated system can be fixed by adding a post-hoc calibration stage at the end of the pipeline, transforming the posteriors output by the system into better posteriors. We believe the only role of calibration metrics should be to aid the developer in deciding whether such post-hoc calibration stage is needed for a given system. After that decision has been made, though, we argue, calibration metrics should play no further role in evaluation. In particular, calibration metrics should not be used to compare systems with each other or to decide whether a system is safe for use in high-stakes applications. EPSRs should instead be used for these purposes.

If the role of calibration metrics is simply to reflect the performance gain that can be achieved by fixing the potential miscalibration of the system, then, we argue, they should be designed to do exactly that. A direct way to measure the impact of a classifier's miscalibration is to add a post-hoc calibration stage to the system and assess the gain in EPSR provided by this addition. The difference between the EPSR of the original classifier and the EPSR after adding post-hoc calibration—called calibration loss—is a direct measure of the miscalibration of the original system. If the calibration loss relative to the EPSR of the original system is large, then a calibration stage should be added to the system. Importantly, the calibration stage should be designed and trained as any other system stage, without using the test data to avoid overestimating the impact that a calibrator would have in the system's performance.

We compared calibration loss with the ECE, the most widely used calibration metric in the machine learning literature, and showed both theoretically and empirically that the ECE has a number of important disadvantages. First, it lacks interpretability since it is not related to a PSR and cannot be turned into a relative measure of the gains that would be achieved with post-hoc calibration. In addition, the multi-class version of ECE measures only the performance of the largest posterior probability generated by the system for each sample rather than of the full posterior, failing to directly address the question of interest. In contrast, calibration loss is directly interpretable, evaluating the calibration of the full posterior distribution as the gain that can be obtained from adding a post-hoc calibrator to the system. Other problems with the ECE include the fact that it relies on histogram binning, an often poor choice as calibration transformation, and that this transformation is trained on the test data itself, leading to potential overestimation of the miscalibration due to overfitting of the transform. While calibration loss could also be computed with this choice of calibration transform, we recommend against it, favoring instead the use of a direction-preserving calibration trained with cross-validation on the test data or on held-out data.

Finally, we also compared calibration loss with a calibration metric obtained from a classic decomposition of the expected PSR of the classifier. We show that, in practice, this decomposition is highly affected by the quality of the calibration transform used to compute it, loosing its meaning when the transform does not adequately fix the classifier's miscalibration. Calibration loss, on the other hand, retains its interpretation as

the gain that can be obtained after post-hoc calibration with the selected transform, even if the calibration transform is suboptimal.

The paper is accompanied by an open-source repository which provides code for the computation of the various metrics in this paper: `https://github.com/luferrer/expected_cost`. All plots and tables in this paper can be replicated using the notebooks under the notebooks/evaluating_posteriors_paper folder in that repository. We hope that this work along with the provided code will facilitate the wider adoption of these metrics that offer a principled, elegant, and general solution to the evaluation and diagnosis of probabilistic classifiers.

### Acknowledgments

This material is based upon work supported by the European Union's Horizon 2020 research and innovation program under grant No 101007666/ESPERANTO/H2020-MSCA-RISE-2020.

We thank Niko Brümmer for many enlightening discussions about calibration and PSRs, for his valuable feedback on this paper, and for contributing the proof in Section 2.4.

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

## A  Cross-entropy and Brier score parameterized by the priors

The cross-entropy (CE) and the Brier scores (BS) can be expressed as a function of the priors, allowing us to turn them into parameters of the metric. For the CE, the expression is given by

$$\text{CE} = -\sum_{t=1}^{N} \frac{P_{h_t}}{N_{h_t}} \log(q_{h_t}(x_t)) \tag{36}$$

where $N_{h_t}$ is the number of samples of class $h_t$. Setting the prior for each class $h$, $P_h$, to $N_h/N$ we recover the standard CE expression (Equation 8). The advantage of the expression above is that it allows us to manipulate $P_h$ independently of the test dataset. This is useful when the class frequencies present in the test data do not reflect the prior distribution that is expected during deployment. In that case, the $P_h$'s can be set to those we expect to see when the system is used. The resulting CE will then better reflect the values we would measure on that target data if it was available for evaluation.

For the BS, the expression parameterized by the priors is given by:

$$\text{BS} = \sum_{t=1}^{N} \frac{P_{h_t}}{N_{h_t}} \frac{1}{K} \sum_{i=1}^{K} (q_i(x_t) - I(h_t = H_i))^2 \tag{37}$$

## B  The absolute distance loss is not a score divergence

Here we provide a counterexample to show that the absolute distance loss does not satisfy the mean-as-minimizer property of Bregman divergences (see Equation (29)). To show this, we generate posteriors $\mathbf{s}$ as described in Appendix C for a 2-class problem with $P_1 = 0.6$. Then, we take the expectation with respect

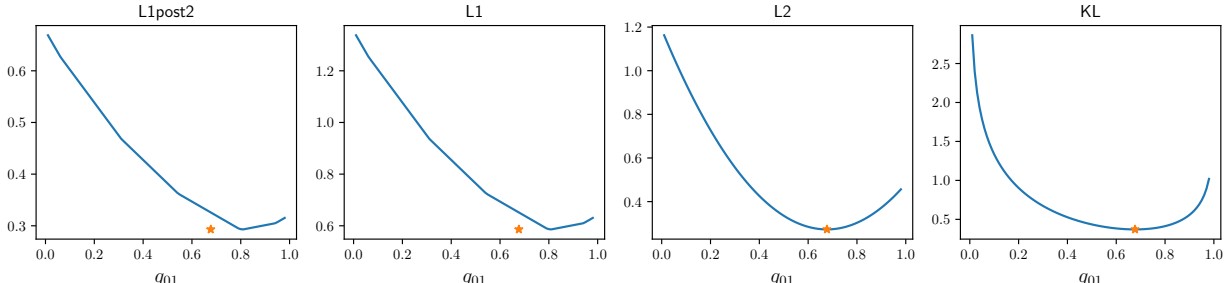

Figure 7: Expected divergence between synthetic posteriors for a binary task and $\mathbf{q}_0$, a fixed vector of posteriors. The figure shows the expected divergence as a function of the first component of $\mathbf{q}_0$, which we call $q_{01}$, for different divergences (blue curves) and the mean value of the first component of the posterior (star). The star has to coincide with the minimum of the curve for valid score divergences proving by contradiction that the L1 loss is not a score divergence.

to the empirical distribution of their divergence to a fixed posterior $\mathbf{q}_0$. For valid score divergences, the minimum expected divergence should occur when $\mathbf{q}_0$ is equal to the mean of $\mathbf{s}$.

Figure 7 shows the expected divergence as a function of the first component of $\mathbf{q}_0$, for four different divergences: 1) the L1 loss only over the posterior for $H_2$, which is, approximately (given that the ECE actually quantizes the posteriors before computing the distance), what the ECE computes, 2) the L1 loss between the full posterior vectors, 3) the L2 loss, which corresponds to the Brier score PSR, and 4) the Kullback-Leibler loss, which corresponds to the negative logarithmic loss PSR. The red stars in this plot correspond to the mean value of the first component of $\mathbf{s}$. We can see that the first two losses do not satisfy the mean-as-minimizer property and, hence, are not score divergences.

## C  Synthetic dataset for experiments

Here we describe the procedure used to create synthetic datasets for the experiments in this paper. Given a number of classes $K$, a total number of samples $N$, a prior for class $H_1$ of $P_1$, and a variance $\sigma$, which are taken as a parameters of the simulation, we proceed as follows:

1. Set the class priors to $P_1$ for class $H_1$, and $P_i = (1 - P_1)/(K - 1)$ for classes 2 through $K$. Unless otherwise indicated, $P_1$ is set to 0.8.

2. Determine the number of samples for each class, $N_i$ as the closest integer to $P_i N$.

3. Generate $N_i$ samples using a multivariate Gaussian distribution $\mathcal{N}(\mu_i, \sigma I)$, with mean $\mu_i$ given by a one-hot vector with the one at the ith dimension and diagonal covariance matrix with equal variance in all dimensions given by $\sigma$ which, unless is otherwise indicated, is set to 0.15. We take these samples to be the $x_t$, the input features for each sample.

4. Compute the likelihoods for each class for each generated sample according to the class distributions used to draw these samples, that is, $P(x|H_i) \sim \mathcal{N}(\mu_i, \sigma I)$.

5. Finally, we assume two possible prior distributions: 1) the same prior distribution given by the $P_i$s according to which the data was generated in step 3, and 2) a mismatched distribution $\hat{P}$, where $\hat{P}_i = 0.1/(K - 1)$ for $i \neq K$ and $\hat{P}_K = 0.9$.

6. Using those two prior distributions, compute two sets of posteriors which we call *cal* and *mcp* (for **m**is-**c**alibrated due to a mismatch in **p**osteriors) which correspond to using the data priors and the mismatched priors, respectively, to obtain the posteriors according to:

$$P(H_i|x) = \frac{P(x|H_i)\ P(H_i)}{P(x)} = \frac{P(x|H_i)\ P(H_i)}{\sum_j P(x|H_j)\ P(H_j)}. \tag{38}$$

| Dataset | #Classes | #Samples | Priors |
|---|---|---|---|
| SST2 | 2 | 1821 | 0.50 0.50 |
| SITW | 2 | 721788 | 0.99 0.01 |
| FVCAUS | 2 | 114072 | 0.98 0.02 |
| CIFAR-1vsO | 2 | 10000 | 0.99 0.01 |
| CIFAR-2vsO | 2 | 10000 | 0.99 0.01 |
| PNEUM | 2 | 624 | 0.38 0.62 |
| ADRENAL | 2 | 298 | 0.77 0.23 |
| PATH | 9 | 7180 | 0.05 0.06 0.08 0.09 0.10 0.12 0.14 0.17 0.19 |
| IEMOCAP | 4 | 5473 | 0.20 0.29 0.31 0.20 |
| AGNEWS | 4 | 7600 | 0.25 0.25 0.25 0.25 |
| CIFAR10 | 10 | 10000 | 0.10 for all classes |

Table 3: Number of classes, number of samples, and class priors for each dataset included in our experiments.

where the $P(x|H_i)$ are the likelihoods computed in step 4 and $P(H_i)$ are the corresponding matched or mismatched priors, $P_i$ and $\hat{P}_i$, respectively.

Note that with the procedure above, the cal posteriors are perfectly calibrated for the test data. The mcp posteriors, though, are not calibrated because, even though the likelihoods used to compute it are obtained from the generating distribution, the priors are mismatched to the ones used for testing. Further, we create misscalibrated versions of the posteriors which we call *mcs* and *mcps* by scaling the cal and mcp posteriors, respectively, in the log domain and then converting them back to posteriors by computing the softmax. The scale is set to 5.0 unless otherwise indicated, simulating an overfitted system that produces overconfident posteriors.

## D   Real datasets for experiments

Table 3 shows the priors and total number of samples for all the datasets used in Section 3.5.

## E   Confidence Intervals

The metrics in the main text and in prior sections of this appendix are computed as averages over the whole test set. For small datasets, this estimate might be quite unreliable, not necessarily reflecting the performance we would observe in practice. Hence, having an estimate of the range of values that our metric of interest could take on a new dataset is essential, specially for small datasets. A standard method for obtaining such ranges is the bootstrapping approach (Tibshirani & Efron, 1993; Keller et al., 2005; Poh & Bengio, 2007). Given a test set of size $N$, the procedure for obtaining confidence intervals is as follows:

- Obtain $B$ bootstrap sets of size $N$ by sampling the original test set with replacement.

- Compute the metric of interest in each of the $B$ bootstrap sets.

- Compute the confidence interval with confidence level $\gamma$ by calculating the $\gamma/2$ and $100 - \gamma/2$ percentiles from the list of $B$ metric values computed above.

The confidence intervals obtained in this way assume that the system is fixed and exactly what will be deployed. Only the variability due to the test data is reflected in these intervals (Raschka, 2018). Importantly, when computing calibration metrics, the calibration transform, if computed using cross-validation or train-on-test rather than using a separate calibration dataset (in which case the calibration transform can be considered part of the system and also frozen), should be retrained for each bootstrap set. This allows the confidence interval to reflect the variability in metric values due to changes in the transform training data. Importantly, when doing cross-validation, the folds should be defined using the original sample identifier to avoid having the same sample across more than one fold.

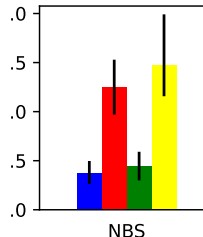 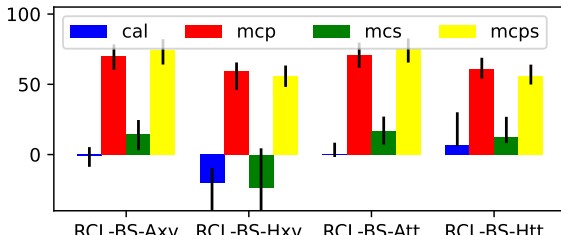

Figure 8: Metrics with confidence intervals obtained with bootstrapping. Left: the normalized BS for the same posteriors as in Figure 5. Right: the RCL for BS using the same calibration methods as in Figure 5.

To illustrate, Figure 8 shows the confidence intervals obtained from 100 bootstrap samples for the same posteriors as in Figure 5 for NBS and BS-based RCL. The code to create this plot can be found in the accompanying repository.

