# OpenReview forum: "Evaluating Posterior Probabilities:  Decision Theory, Proper Scoring Rules, and Calibration"
_TMLR — Accepted by TMLR_

### Review · Reviewer_wX9M · 2024-10-06

**Summary Of Contributions:**

This paper discusses the evaluation of probabilistic predictions. It focuses on proper scoring rules (PSRs) and various notions of calibration. The paper aims to persuade readers that calibration should not be relied on to evaluate systems. The bulk of the theoretical presentation is devoted to prior work. There are experiments on synthetic and real examples. The authors introduce a new notion of "calibration loss," which quantifies how much a scoring rule improves when a classifier is post-processed to be calibrated. This notion generalizes an older definition (for binary classification) that appears to not be widely used.

**Audience:**

Yes

**Broader Impact Concerns:**

None.

**Claims And Evidence:**

Yes

**Requested Changes:**

There are several different notions of multiclass calibration that would be useful to point to, even if they don't change the discussion. There is probably a better reference, but [1] is a recent paper with relevant citations.

I think the paper would be much better if the authors were able to supply a direct argument for why calibration is not a sensible desideratum. I view the current argument, that PSRs lead to a single number which is more meaningful and useful, as indirect.

[1] Gopalan, P., Hu, L. & Rothblum, G.N.. (2024). On Computationally Efficient Multi-Class Calibration. Proceedings of Thirty Seventh Conference on Learning Theory, in Proceedings of Machine Learning Research 247:1983-2026 Available from https://proceedings.mlr.press/v247/gopalan24a.html.

**Strengths And Weaknesses:**

The authors of this paper clearly communicate their goals and I found it easy to understand their logic. They do a good job building up their technical tools in a cohesive fashion. One aspect I appreciated was the explicit use of a "reference distribution" in place of the questionable "true" distribution.

My main issue with the paper is that I am unpersuaded by it. The authors' position is that, because PSRs exactly capture a probabilistic classifier's quality under a cost function, it doesn't make sense to care about calibration. However, I take it as self-evident that calibration is a desirable property. So one might want a system with both a low score and high calibration. It seems reasonable that, in some settings, one might be willing to sacrifice a little Brier score for much better calibration. Thus, I feel the paper contains many statements which are not supported by the arguments. For example: "...assessing calibration is not helpful to determine the quality of posterior probabilities and should never be used to compare systems with each other." (pages 20 & 22).

Most readers will understand such statements as expressing opinions, so I don't view this as an issue of technical soundness.

Overall, while this piece doesn't supply much in the way of new findings or new concepts, I think that it contributes to the discussion in a way that some will find interesting.

---

> ### Author Response · Authors · 2024-10-17
> **Discussion on our main claim**
>
> Dear Reviewer,
>
> We thank you for your thoughtful review which raises an interesting discussion. We will try to further explain the claim that calibration is not a valid metric to assess the quality of a system that produces posterior probabilities.
>
> We do not see the argument as an opinion, as it derives directly from the definition of calibration and two simple premises:
>
> * Users of a classification system are interested in getting good posterior probabilities for the classes *given the input sample*, that is, given all the information available in that sample. They wish to know how likely each class is for that sample.
>
> * Performance metrics should reflect as directly as possible the needs of the user. Hence, in the scenario of interest, the metric should reflect the quality of the posteriors
> probabilities for the classes *given the input sample*.
>
> Now, by definition, calibration does not deal with those posteriors but rather with the posteriors for the classes *given the scores output by the system* which are *not* the posterior that the user cares about. In particular, if the system is poor, the output scores may have a very tenuous or even nonexistent relationship with the input sample. In the extreme case, a classifier could ignore the input sample completely and produce always the same output. If that output was given by the prior probability of the classes, then the classifier would be perfectly calibrated, but it would be of no value to the user. A calibration metric would completely fail to diagnose the fact that the system is useless. In general, since calibration metrics assess the quality of a posterior probability that is of no particular interest to the user, they should not be used to assess system performance.
>
> On the other hand, calibration analysis is quite important during development. Good calibration is indeed a desirable property because miscalibration implies that the system can be improved by adding a post-hoc calibration stage. Hence, during development it is wise to assess whether the system is well calibrated, just as it is important to select the best system architecture and the best hyperparameters. Like learning curves are useful to detect overfitting, calibration analysis is useful to decide whether the system requires a post-hoc calibration stage. Yet, just as whether the system overfitted or underfitted the training data is of no concern to end users, whether the system is calibrated or not is also not relevant to them. *Fixing miscalibration is one way to improve the quality of the posteriors, but calibration performance is not a reflection of the resulting quality.*
>
> In other words, the calibration performance of a system can be seen as just one of its many characteristics, like the number of parameters, the type of architecture, or the data it was trained with. Those are all aspects that *affect* but do not *reflect* the performance of the system.
>
> Further, calibration is not like run-time or memory consumption or explainability, which are often valuable additional objectives to trade-off against performance. Users should never pick a system with a lower calibration loss over a system with a lower EPSR. They would simply be choosing a system with worse posterior probabilities *given the input samples* – a poorer system for decision-making – without any advantage. Having a well-calibrated system has no impact on how useful the posteriors are for interpretation or decision making. So, while it is sometimes necessary to choose a system that has slightly worse performance than another one but much smaller computational requirements, we see no reason to choose a system that has slightly worse EPSR but better calibration.
>
> Please, let us know what you think about these arguments. We are very happy to continue this discussion which has important practical implications.
>
> With respect to the various notions of calibration, we will add a note explaining that the paper deals with canonical calibration (sometimes also called strong calibration, as in [1]), except when explaining ECE, which assesses calibration of the confidence.
>
> [1] D. Widmann, F. Lindsten, and D. Zachariah. Calibration tests in multi-class classification: A unifying framework. In Proc. of NeurIPS, Vancouver, December 2019

---

> > ### Comment · Reviewer_wX9M · 2024-10-17
> >
> > Thank you for the response, which I view as a (very good) summary of the submission's main points. It did not cause me to significantly update my beliefs.
> >
> > I think your comparison with running time and memory consumption exactly capture our disagreement. It seems to me that some person, let's call him Businessman Bob, might (i) have a strong preference for simple metrics and (ii) believe that calibration is "simple." Bob finds scoring rules unintuitive. So he prefers highly-calibrated Classifier A, even though his engineers tell him that Classifier B, which is much less calibrated, has a higher EPSR. Your paper argues that Bob is making the wrong choice.
> >
> > However, I don't know how one would mathematically convince Businessman Bob to change his mind. That's why I refer to claims such as "calibration metrics... should not be used to assess system performance" as "opinions," since I don't see how to turn them into facts. I don't mean the word in any pejorative sense.

---

> > > ### Author Response · Authors · 2024-10-21
> > > **Rational decision making**
> > >
> > > Thank you again for posing this interesting and nuanced discussion in such clear terms.
> > >
> > > Indeed, we would argue that Bob is making the wrong choice. More concretely, he is making a suboptimal choice in terms of cost. If he chooses a system based on a calibration metric he could end up making suboptimal decisions which could result, for example, in less sales and less income for his company. Unlike for running time or memory consumptions which are real constraints Bob might face in practice, he is simply choosing a worse system without any actual benefit. We would argue that Bob is being irrational. If we were his engineers, we would try to convince him by explaining that by making irrational decisions he will end up losing money. If Bob is a good businessman, he would probably be persuaded by this argument.
> > >
> > > Our underlying premise in this work is that decisions should be made rationally and that, under uncertainty, rational decisions are supported by statistical decision theory. This is a widely accepted view in the literature on decision-making (see, for example in [1,2]), but it may still be seen as an opinion. So, we agree with you, the main claim in the paper is founded on an opinion. Yet, note that the statement that calibration metrics should not be used to assess performance is not an additional independent opinion but rather, derives directly from that one fundamental premise. We will include an explicit discussion of this assumption in the introduction.
> > >
> > > [1] I. Good. Rational decisions. Journal of the Royal Statistical Society, 14, 1952.
> > >
> > > [2] M. Peterson. An Introduction to Decision Theory. Cambridge Introductions to Philosophy. Cambridge University Press, 2009.

---

### Review · Reviewer_Qdxh · 2024-10-21

**Summary Of Contributions:**

This paper advocates against an over-reliance on calibration metrics to evaluate probabilistic classifiers. To do so, the paper first provides a lengthy tutorial and discussion on Bayesian decision theory, proper scoring rules, and calibration. The paper then presents an argument that "the goodness of a probabilistic system...should be assessed using [expected proper scoring rules], not calibration metrics" and further advocates for a very restricted use of calibration metrics and the use of one calibration metric in particular.

**Audience:**

Yes

**Claims And Evidence:**

No

**Requested Changes:**

The paper needs to be much clearer about what exactly its stance on calibration is and how it is actually different than the dominant viewpoint in machine learning. I am not aware of a large community in ML that actually believes the calibration of a probabilistic classifier equates to its goodness, regardless of its discriminative capability. I am not sure if the paper is making that claim, but if it is, it needs to substantiate it much more convincingly. My current understanding of this paper is that it is a position paper which is pointing out and correcting a currently widespread bad practice in the ML community. For this to be a useful paper, the bad practice needs to be more clearly stated, and its widespread-ness needs to be more convincingly shown.

**Strengths And Weaknesses:**

I'm keeping my comments short, because I think there are some fundamental issues with the submission. If the authors respond convincingly to these issues, I will take another look at the paper and provide a more thorough review.

The paper is essentially a position paper advocating against equating the calibration of a probabilistic classifier with its goodness, which the paper represents as a widespread viewpoint/practice in machine learning. The paper is certainly correct to point out that "calibration is neither necessary, nor sufficient for posterior probabilities to be useful to the end user". However, I believe (and am not convinced otherwise by the paper) that this is already the dominant viewpoint in machine learning, and that there are relatively few papers actually arguing otherwise.

The paper cites a small handful of papers repeatedly in making its central claim that there is an over-reliance on calibration metrics in the ML literature. For instance, the paper says "many of the recent machine learning works concerned with the evaluation of posteriors do not use PSRs for the task, resorting instead to calibration metrics (for example, Guo et al., 2017; [...])". This statement surprised me, so I took a look at the first citation, Guo et al. (2017), to get a sense for how they were using calibration metrics. Guo et al. (2017) seem to use such metrics as a post-processing step to calibrate the outputted probabilities of a trained neural network. Importantly, the neural networks in their paper are first trained to minimize cross-entropy loss (a proper scoring rule). In what sense then can it be said that Guo et al. do not use a PSR for the evaluation of posteriors when their models are explicitly trained to perform well on a PSR?

The paper's actual stance on calibration metrics is confusing and difficult to pin down, and the paper sometimes seems to contradict itself. The paper, for instance, makes the following strong claim:

1) "Hence, when evaluating the utility of posteriors we should not be concerned with whether they are calibrated or not."

which is followed shortly by this one:

2) "In this paper, we argue that the only purpose of calibration metrics should be to diagnose whether a system is well-calibrated in order to fix it if that is not the case [...]"

If we should not be concerned with whether posteriors are calibrated (claim 1), then why should we care about fixing systems that are not well-calibrated (claim 2)? Moreover, how does the practice that the paper advocates in claim 2 differ meaningfully from that of Guo et al. (2017) (who use calibration metrics to diagnose whether fitted neural networks are not well-calibrated and to fix them if so)?

---

> ### Author Response · Authors · 2024-10-23
> **Response - part 1**
>
> Thank you very much for your questions and comments.
>
> Let us first respond to the comment about the apparently contradictory claims. The first claim states that calibration metrics should not be used to evaluate the performance of probabilistic classifiers. That is, they should not be used to decide which system is better or to determine whether a system is good enough for a certain application. This claim derives from the fact that good calibration is neither necessary nor sufficient for a system to produce good posteriors. Hence, calibration metrics are not performance metrics. In essence, using calibration metrics as performance metrics is like using the number of layers in a DNN as a performance metric. While both miscalibration and the number of layers *affect* the performance of such DNN, they do not *measure* it.
>
> The second claim, on the other hand, states that calibration metrics may be used during development to decide whether a post-hoc calibration stage can improve the performance of the system, just like learning curves may be used to diagnose overfitting. Bad calibration degrades the performance of the posteriors, so it is always a good idea during the development process to determine whether the system is well calibrated and, if it is not, to fix the problem by adding a calibration stage. Yet, while calibration metrics are a useful diagnostic tool during development, they should not be used to measure the performance of the resulting system.
>
> Guo's paper does indeed use calibration metrics as a way to diagnose calibration problems (in an indirect way since they use one approach – histogram binning – to diagnose the problem, and a different approach – temp scaling – to fix it, but that is a side issue that is not central to this discussion). Our main issue with Guo's paper is their statements on the value of calibration as a *requirement* for systems that are used for real-world decision making. For example, they state that "In real-world decision making systems, classification networks must not only be accurate, but also should indicate when they are likely to be incorrect. ... Specifically, a network should provide a calibrated confidence measure in addition to its prediction. … Calibrated confidence estimates are also important for model interpretability." Further, the paper states that "NLL (negative log-likelihood) can be used to indirectly measure model calibration," further indicating that their goal is to obtain better calibration, rather than a better EPSR, which would be the correct goal.
>
> After Guo's paper, a large number of subsequent works include similar statements. Below are some example quotes from other highly cited papers:
>
> * "In safety-critical applications a probabilistic model is usually required to be calibrated, i.e., to capture the uncertainty of its predictions accurately." [Widmann19]
> * "Calibration is an essential component of the evaluation of computational models for medical decision making, diagnosis, and prognosis." [Huan20]
> * "Finally, we performed extensive experiments on how label smoothing can implicitly calibrate model’s predictions. This has big impact on model interpretability ..." [Muller19]
> * "Miscalibration ... of Deep Neural Networks (DNNs) makes their predictions hard to rely on." [Mukhoti20]
> "Applications such as weather forecasting and personalized medicine demand models that output calibrated probability estimates---those representative of the true likelihood of a prediction." [Kumar19].
> * "Accurate estimation of predictive uncertainty (model calibration) is essential for the safe application of neural networks." [Minderer21].
> * "The reliability of a machine learning model’s confidence in its predictions is critical for high-risk applications. Calibration ... formalizes this notion." [Nixon19]
> * "In safety-critical applications, it is pivotal for a model to possess an adequate sense of uncertainty, which for probabilistic classifiers translates into outputting probability distributions that are consistent with the empirical frequencies observed from realized outcomes. A classifier with such a property is called calibrated." [Vaicenavicius19]
> * "It is also crucial to assure that the uncertainty estimates are reliable. To this end, the calibration property (the degree of reliability) of DNNs has been investigated and re-calibration methods have been proposed to obtain reliable (well-calibrated) uncertainty estimates" [Gawlikowski23]
>
> As Guo's work, these papers give calibration a special status, stating or implying that a system is only safe, reliable or interpretable if it is well calibrated. Calibration metrics are then believed to be the natural tool for evaluating the quality of the posteriors.

---

> > ### Author Response · Authors · 2024-10-23
> > **Response - part 2**
> >
> > Note that we are not saying that the community "believes the calibration of a probabilistic classifier equates to its goodness, *regardless of its discriminative capability*". Papers do acknowledge that calibration alone is not enough to assess a system's performance. We are saying that the community believes that good calibration is an essential characteristic for posteriors to be good and we argue that it is not and, hence, that calibration metrics have no role to play in system evaluation – not even if accompanied by a discrimination-sensitive metric. As mentioned above: we see them only as a diagnostic tool to detect calibration problems during system development.
> >
> > Some of the papers that report calibration metrics use them, as in Guo's paper, to diagnose miscalibration, which is fine with us (except we would not use ECE but calibration loss for that purpose). Yet, they then also use them to show which post-hoc calibration approach is better at reducing the miscalibration of the system. In practice, this is somewhat equivalent to reporting an EPSR. Since post-hoc calibration does not change the discrimination of the system, an improvement in the calibration metric implies an improvement in the quality of the posteriors. Hence, in these cases, reporting calibration metrics is, in fact, an indirect way of reporting what the user really cares about, which is whether the quality of the posteriors was improved by the post-hoc calibration stage. We argue that it would be conceptually and practically simpler and more informative to report the value of an expected PSR, a metric designed to directly assess the quality of the posteriors.
> >
> > More concerning, though, is the wide-spread practice of comparing the performance of systems with each other in terms of calibration performance *even when these systems have different discrimination performance*  (see, for example, Table 3 in [VanHoorde15], Figure 2-4 in [Minderer21], Figure 4 and Table 1 in [Mukhoti20], Table 3 in [Muller19], Tables 4-8 in [Jian21], Table 3 in [Desai20], Table 9 in [Dehghani23]). As those papers acknowledge that calibration metrics are not enough to describe the performance of the system, they sometimes also report a discrimination-sensitive metric like classification accuracy. This then results in unnecessary conflict, since an improvement in classification accuracy is often accompanied by a degradation in calibration. Readers are then left wondering which system is actually better, caught in a needless trade off between metrics. This problem would be trivially solved by reporting an expected PSR: the system with a better EPSR has better posteriors, regardless of the quality of its calibration.*
> >
> > The papers above are only some examples of the bad practice we aim to rectify: reporting calibration metrics to assess the quality of posteriors, justified by a belief that good calibration is a requirement for posteriors to be safe, reliable or interpretable. Many more examples could be listed. In fact, in current machine learning literature, most papers concerned with the quality of posteriors report calibration metrics, citing Guo's work and motivating this practice with statements along the same lines. This is a concerning trend that we hope our paper will help to revert.
> >
> > *When decisions are made using Bayes decision theory, the error rate (which is one minus the classification accuracy) is, in fact, an EPSR. Yet, this metric does not adequately assess the quality of the posteriors across the whole simplex since it focuses on a single operating point. To this end, the expectation of a strict PSR like the cross-entropy or the Brier score are needed.

---

> > > ### Author Response · Authors · 2024-10-23
> > > **Response - part 3 (references)**
> > >
> > > **References with number of citations**
> > >
> > > Guo et al., "On Calibration of Modern Neural Networks", PMLR, 2017. Over 6200 citations
> > >
> > > Widmann et al., "Calibration tests in multi-class classification: A unifying framework", NeurIPS, 2019. 111 citations
> > >
> > > Huan et al., "A tutorial on calibration measurements and calibration models for clinical prediction models", Journal of the american medical informatics association, 2020. 286 citations
> > >
> > > Müller et al., "When does label smoothing help?", NeurIPS, 2019. 2277 citations
> > >
> > > Mukhoti et al., "Calibrating deep neural networks using focal loss", NeurIPS 2020. 456 citations
> > >
> > > Kumar et al., "Verified uncertainty calibration", NeurIPS, 2019. 378 citations
> > >
> > > Minderer et al., "Revisiting the Calibration of Modern Neural Networks", NeurIPS, 2021. 327 citations
> > >
> > > Nixon et al., "Measuring calibration in deep learning", CVPR workshop, 2019. 488 citations
> > >
> > > Vaicenavicius et al., Evaluating model calibration in classification. PMLR, 2019. 244 citations.
> > >
> > > Gawlikowski et al., "A survey of uncertainty in deep neural networks." Artificial Intelligence Review , 2023. 1091 citations
> > >
> > > Van Hoorde et al., A spline-based tool to assess and visualize the calibration of multiclass risk predictions. Journal of biomedical informatics, 2015. 78 citations
> > >
> > > Jiang, et al. "How can we know when language models know? on the calibration of language models for question answering." Transactions of the Association for Computational Linguistics, 2021. 306 citations
> > >
> > > Desai and Durrett. "Calibration of pre-trained transformers." arXiv preprint arXiv:2003.07892, 2020. 259 citations.
> > >
> > > Dehghani et al. "Scaling vision transformers to 22 billion parameters." International Conference on Machine Learning. PMLR, 2023. 411 citations

---

> > > ### Comment · Reviewer_Qdxh · 2024-10-24
> > >
> > > Thanks very much for your detailed response. I want to focus in on a very specific confusion of mine about your position.
> > >
> > > You say:
> > >
> > > "We are saying that the community believes that good calibration is an essential characteristic for posteriors to be good and we argue that it is not and, hence, that calibration metrics have no role to play in system evaluation [...]"
> > >
> > > Then you say:
> > >
> > > "As mentioned above: we see them only as a diagnostic tool to detect calibration problems during system development."
> > >
> > > My question: if calibration is not an essential characteristic for posteriors to be good, then why should calibration ever be diagnosed during system development?
> > >
> > > My current opinion is that:
> > >
> > > EITHER calibration is sometimes an important characteristic for a posterior to be good, in which case it seems right to both diagnose calibration during development and also to diagnose calibration during evaluation...
> > >
> > > ...OR calibration is not something that is important for a posterior to be good, in which case it seems irrelevant to diagnose calibration during development and during evaluation.
> > >
> > > I am still not understanding why it would be ever be right to diagnose calibration during development but not during evaluation. Can you please elaborate?
> > >
> > > Thanks in advance.

---

> > > > ### Author Response · Authors · 2024-10-25
> > > > **Diagnosis vs evaluation**
> > > >
> > > > Thank you for laying out the question so clearly.
> > > >
> > > > First, let us clarify what we mean by diagnosis and evaluation. By diagnosis we refer to the process of figuring out whether and how a system can be improved. If we are doing diagnosis it means we have an intention of going back to the system and changing something to improve it as a consequence of the results of the diagnosis. So, diagnostic tools should be used on development (also sometimes called validation) data, while we are still trying to design the best possible system.
> > > >
> > > > Evaluation, on the other hand, means estimating how a system will perform from a user's point of view. The metrics used for evaluation should reflect as closely as possible the needs of the user. They should be a numerical representation of how useful the system is in practice. While diagnosis only makes sense during development, evaluation is done both during development (to make development decisions and have checkpoints on how well the development process is going, keeping in mind that the performance on that set is likely optimistic) and, after development, using a dataset that was unseen during the development process. This latter evaluation is an estimation of the performance we expect the system to have during deployment.
> > > >
> > > > So, why is calibration not a measure of system performance? Maybe the key to understanding this is to see that the actual measure of the quality of posteriors – the EPSR – can be decomposed in two components, one of which is the calibration loss (see, for example, deGroot & Fienberg, 1983, cited in our paper):
> > > >
> > > > EPSR = calibration_loss + discrimination_loss
> > > >
> > > > Either one of those components on their own is a very poor indication of posterior quality.
> > > >
> > > > In particular, calibration can be very good and the EPSR still be quite large (i.e., poor). For example, a system that outputs the class priors for every single sample has perfect calibration but a high value for the EPSR, because the discrimination loss is as large as it can get. Such a system is essentially useless, providing no information about the input sample. It would be as if a weather forecaster always gave a chance of rain of X% where X is the percentage of days that it rains every year, ignoring any actual data the sensors may be providing. Users would very quickly learn to ignore such forecast, even though it is perfectly calibrated.
> > > >
> > > > Further, calibration can be relatively poor and yet the EPSR can be low enough for the system to be useful, potentially more useful than other perfectly-calibrated systems. Imagine, for instance, that we have two systems. System 1 has an EPSR of 0.4, with a calibration loss of 0.2 and a discrimination loss of 0.2 and System 2 has an EPSR of 0.7 with a calibration loss of 0.0 and a discrimination loss of 0.7. While system 2 is perfectly calibrated, system 1 has better posteriors. One should always choose system 1 over system 2 since it is a more useful system.
> > > >
> > > > Hence, by looking at the calibration term we cannot tell whether a system is good or bad on its own, or better or worse than another system. The calibration loss does not serve as a measure of the goodness of the posteriors. What matters is the sum of both terms.
> > > >
> > > > Yet, while the decomposition is not needed or even useful for knowing how good our system is, it is quite effective as a diagnostic tool. Say we are getting a concerningly large EPSR value. By finding the calibration/discrimination decomposition we can figure out what is wrong with our system. If we find the calibration term is large, we know we can add a post-hoc calibration stage to obtain a lower EPSR (that, if we succeed, will be close to the discrimination term). Otherwise, if the calibration term is small compared to the discrimination term, we are in a more complicated scenario where we need to improve our classifier by going back to the drawing board, trying other architectures, gathering more data, and/or tuning hyperparameters.
> > > >
> > > > The calibration/discrimination decomposition is comparable to many other diagnostic procedures we use during system development. For example, we may probe the activations inside a neural network to see if too many of them have constant values in which case we might attempt to decrease the size of the model. Or we might look at score distributions to find outliers or strange peaks that might suggest issues with our data. We might look at the training loss across batches to assess whether we need to decrease or increase the learning rate or the batch size. We do not use any of those tools to assess the goodness of our systems, but as a guide on how to improve it. The calibration/discrimination decomposition is just one more tool in that toolkit.

---

> > > > > ### Author Response · Authors · 2024-10-25
> > > > > **Diagnosis vs evaluation - summary**
> > > > >
> > > > > In summary: good calibration is not necessary nor sufficient for posteriors to be good (hence, calibration loss does not make sense as a performance metric because it does not measure anything the end user cares about) BUT if a system is miscalibrated, and we are still in the development stage, we can improve it by adding a post-hoc calibration stage (hence, calibration loss is a useful diagnostic tool).
> > > > >
> > > > > We hope this helped clarify our claim. Otherwise, please let us know.

---

> > > > > > ### Comment · Reviewer_Qdxh · 2024-10-26
> > > > > >
> > > > > > I remain confused.
> > > > > >
> > > > > > If calibration “does not measure anything the end user cares about”, then how can you “improve” the posterior by calibrating it (during development)?
> > > > > >
> > > > > > More generally, how can you “improve” a system by altering a property that supposedly does not affect the “good”-ness of the system?
> > > > > >
> > > > > > This is a philosophical point, but yours is a fairly philosophical paper, and this point lies at its core.

---

> > > > > > > ### Author Response · Authors · 2024-10-27
> > > > > > >
> > > > > > > Thank you for your question. We agree, this is an important point to clarify.
> > > > > > >
> > > > > > > Our argument is: calibration does *affect* the goodness of the posteriors but it does not adequately *reflect* it. Because it affects it, it can be used for diagnosis. Because it does not reflect it, it should not be used as a performance metric.
> > > > > > >
> > > > > > > In principle, anything that affects the system performance could potentially be used as performance metric. Yet, one aims for a metric that reflects the goodness of the system as well as possible, a value that is tightly related with how useful the system is for an end user. As we explained in our prior answer, calibration metrics do not satisfy this requirement. While poor calibration implies that the posteriors can be improved it does not mean that they are poor – they are just poorer than they could be if we added a post-hoc calibration to the system.
> > > > > > >
> > > > > > > A good analogy, perhaps, is the size of the training dataset. Increasing the size of the training set can almost always improve performance of a given system. Yet, we do not use that size as a performance metric because it is only a partial indication of the goodness of the system. The same thing applies to calibration.
> > > > > > >
> > > > > > > Please, let us know if this helped.

---

> > > > > > > > ### Author Response · Authors · 2024-10-27
> > > > > > > >
> > > > > > > > PS: The analogy between calibration and training data size goes quite deep:
> > > > > > > >
> > > > > > > > * Knowing the size of the training dataset or its calibration quality does not tell us whether a system is good or bad. A system may be trained with a small amount of data or be miscalibrated and still be quite good. A system may be trained with a large amount of data and be perfectly calibrated and still be poor.
> > > > > > > >
> > > > > > > > * If we increase the size of the training set for a given system, the quality of the system will usually improve (or, at least, not degrade). Similarly, if we calibrate a given system, it will usually improve (or, at least, not degrade).
> > > > > > > >
> > > > > > > > * If we have two different systems, the size of the training set or its calibration quality do not tell us which system is better. A system may be trained with more data or be better calibrated than another but be worse in terms of its performance.
> > > > > > > >
> > > > > > > > Hence, just like the training data size is never used as a measure of system performance, calibration should not either.

---

> > > > > > > > > ### Comment · Reviewer_Qdxh · 2024-10-29
> > > > > > > > >
> > > > > > > > > Thank you, the "reflect versus affect" clarification and the training set analogy are very helpful. The draft should be revised to make this clearer. I need to re-read the paper with this framing in mind, and may return with more questions/comments.

---

> > > > > > > > > > ### Author Response · Authors · 2024-10-29
> > > > > > > > > >
> > > > > > > > > > We are glad that helped. We will add that explanation to the introduction.
> > > > > > > > > >
> > > > > > > > > > We look forward to any further comments or questions.

---

### Review · Reviewer_Jshr · 2024-10-21

**Summary Of Contributions:**

The contributions of the paper are as follows:
* The authors advocate for the use of expected proper scoring rules (PSRs) over calibration metrics like the expected calibration error (ECE) for assessing the quality of posterior probabilities in classifiers.
* The authors provide a practical review of PSRs, grounded in Bayes decision theory, to demonstrate why expected PSRs serve as a principled measure of a system's posterior quality, unlike calibration metrics.
* The authors introduce a new calibration metric called calibration loss, derived from a decomposition of expected PSRs, and show that it is superior to existing metrics

**Audience:**

Yes

**Broader Impact Concerns:**

I don't think there are any broader impact concerns.

**Claims And Evidence:**

Yes

**Requested Changes:**

The only minor requested changes I have are:
1. The inclusion of a greater number of diagrams in the theory section.
2. An increase in the size of the diagrams in the experimental section.
3. Bringing brief explanations of datasets into the main text.

**Strengths And Weaknesses:**

I like this paper. In particular, I believe the following aspects of the paper are a strength:
1. *Discussion of PSRs.* The paper provides a well-rounded review of proper scoring rules (PSRs), explaining their connection to Bayes decision theory. It explains why PSRs are more appropriate for evaluating the quality of posterior probabilities than calibration metrics, which only address one aspect of posterior quality. I like the pedagogy here.
2. *Novel proposition of calibration loss.* I like the grounding and simplicity of this idea. It seems reasonable that one should simply "add a calibration transform at the output of the system, and assess the level of improvement obtained from this additional stage" to determine the miscalibration.
3. *Critique of current practices.* The paper critiques common practices in machine learning where calibration metrics (like ECE) are used to assess posterior performance without considering discrimination or refinement aspects. This critique seems timely and valuable.

The only weakness in my mind is the relative lack of practical experiments. An exploration of real-world, high-stakes applications (e.g., medical diagnostics, autonomous vehicles) would strengthen the practical relevance of the proposed calibration loss. However, this kind of critique can almost always be used against authors, so I do not weight it highly.

---

> ### Author Response · Authors · 2024-10-24
> **Response to review**
>
> Thank you very much for your review. We are glad you liked the paper. We will work on the required changes and submit a revised version as soon as possible.
>
> With respect to the experimental results on high-stakes applications, the speaker verification tasks in section 3.5 (SITW and FVCAUS) can be considered high-stakes since that technology is used for biometric access to sensitive information like bank accounts and for forensic biometrics, among many other applications. The results on those tasks illustrate the advantage of the calibration loss over the ECE, which fails to diagnose the extreme miscalibration of some of the systems. Yet, we agree that it would be interesting to include further results on some other high-stakes application. We have run the evaluation on the Medical imaging tasks in MedMNIST and we will include some of those results for further illustration of our claims.

---

### Decision · Action_Editor_Vn66 · 2024-12-24

**Recommendation:** Accept as is

**Comment:**

After the rebuttal and discussions, the 3 reviewers agreed with accepting the paper, by recommending accept or leaning accept.

Reviewer wX9M requested the following in the review: "There are several different notions of multiclass calibration that would be useful to point to, even if they don't change the discussion. There is probably a better reference, but [1] is a recent paper with relevant citations."
[1] https://proceedings.mlr.press/v247/gopalan24a.html
I encourage the authors to look into the reference (and relevant citations) to further enhance the discussions about multi-class calibration.

**Audience:**

One reviewer pointed out that if there is "an epidemic of misunderstanding in the ML community of this material, then such a paper is warranted." However, the reviewer is "not sufficiently aware of the sub-communities mentioned in this article to assess whether such misunderstanding/malpractice really is so widespread." Another reviewer mentioned that "This is essentially a position paper, but some readers may its perspective interesting." I agree that some individuals in TMLR's audience would be interested in knowing the content of this paper, since many researchers use proper scoring rules or calibration metrics.

**Claims And Evidence:**

The paper first provides a tutorial and discussion on Bayesian decision theory, proper scoring rules, (PSRs; such as the Bayes risk, cross-entropy, and Brier score), and calibration metrics such as the expected score divergence and expected calibration error (ECE). The paper advocates the use of PSRs over calibration metrics such as ECE. The paper proposes a metric called the calibration loss. The experiments demonstrate the properties discussed in the paper. The reviewers had a few concerns about the claims and position of the paper, but after the rebuttal and discussions, all reviewers were satisfied with the content of the paper. The authors have provided a revision that addressed the comments and suggestions provided by the reviewers.